# A cancer rainbow mouse for visualizing the functional genomics of oncogenic clonal expansion

Peter G. Boone[1,2,9,10], Lauren K. Rochelle[1,2,9,10], Joshua D. Ginzel[2], Veronica Lubkov[1,2], Wendy L. Roberts[2], P.J. Nicholls[2], Cheryl Bock[3], Mei Lang Flowers[3], Richard J. von Furstenberg[4], Barry R. Stripp[5], Pankaj Agarwal[1], Alexander D. Borowsky[6], Robert D. Cardiff[6], Larry S. Barak[2,10], Marc G. Caron[2,10], H. Kim Lyerly[1,7,8,10] & Joshua C. Snyder [1,2,10]*

Field cancerization is a premalignant process marked by clones of oncogenic mutations spreading through the epithelium. The timescales of intestinal field cancerization can be variable and the mechanisms driving the rapid spread of oncogenic clones are unknown. Here we use a Cancer rainbow (Crainbow) modelling system for fluorescently barcoding somatic mutations and directly visualizing the clonal expansion and spread of oncogenes. Crainbow shows that mutations of ß-catenin (*Ctnnb1*) within the intestinal stem cell results in widespread expansion of oncogenes during perinatal development but not in adults. In contrast, mutations that extrinsically disrupt the stem cell microenvironment can spread in adult intestine without delay. We observe the rapid spread of premalignant clones in Crainbow mice expressing oncogenic Rspondin-3 (*RSPO3*), which occurs by increasing crypt fission and inhibiting crypt fixation. Crainbow modelling provides insight into how somatic mutations rapidly spread and a plausible mechanism for predetermining the intratumor heterogeneity found in colon cancers.

[1] Division of Surgical Sciences, Department of Surgery, Duke University School of Medicine, Durham, NC, USA. [2] Department of Cell Biology, Duke University School of Medicine, Durham, NC, USA. [3] Transgenic Mouse Facility, Duke Cancer Institute, Durham, NC, USA. [4] Division of Gastroenterology, Department of Medicine, Duke University School of Medicine, Durham, NC, USA. [5] Department of Medicine and Biomedical Sciences, Cedars-Sinai Medical Center, Los Angeles, CA, USA. [6] Department of Pathology and Laboratory Medicine and The Center for Comparative Medicine, University of California-Davis, Davis, CA, USA. [7] Department of Pathology, Duke University School of Medicine, Durham, NC, USA. [8] Department of Immunology, Duke University School of Medicine, Durham, NC, USA. [9] These authors contributed equally: Peter G. Boone, Lauren K. Rochelle. [10] These authors jointly supervised this work: Larry S. Barak, Marc G. Caron, H. Kim Lyerly, Joshua C. Snyder. *email: joshua.snyder@duke.edu

The process of field cancerization displaces normal cells with fields of premalignant cells harboring mutations to proto-oncogenes and tumor suppressors. Field cancers are operationally defined by Graham and colleagues as "a group of cells that are considered to be further along an evolutionary path toward cancer"[1]. Fields can present with or without morphologic pathology. The initiation and spread of field cancers increase the risk of a tumorigenic event setting the stage for sporadic cancers[1–3]. Determining how field cancerization occurs may lead to new strategies for cancer surveillance and treatment. The extent of field cancerization in the intestine varies from a few centimeters adjacent to a tumor or, in other circumstances, can include nearly the entire colonic epithelium[4,5]. The current model of intestinal field cancerization begins with the acquisition of a somatic mutation in an intestinal stem cell (ISC) within a crypt. Somatic mutations increase the competitive fitness of the ISC, resulting in the replacement of neighboring wild-type (WT) ISCs and the fixation of the mutation in a single crypt[6]. Premalignant crypts then duplicate by crypt fission to outcompete and displace adjacent normal crypts[7–9] (Fig. 1a). The entire process is thought to occur slowly over several decades[7,10].

One reason for the proposed slow progression of field cancers is the paradoxical finding that premalignant ISCs with somatic mutations, such as Kras[G12D], APC-loss, or mutant TP53, can be stochastically replaced by neighboring WT ISCs. Therefore, field cancers can be prematurely extinguished by the healthy intestine[10]. A second reason for the proposed slow progression of field cancers is that healthy adult intestinal crypts infrequently duplicate—a process termed crypt fission. Less than 2% of crypts are undergoing fission in adults. Each crypt may only undergo one fission event every 30–40 years in the healthy intestine[9,11]. Therefore, the spread of field cancers is also severely limited. Crypt fission can be increased by somatic mutations. However, in familial adenomatous polyposis (FAP) patients and in mouse models of APC inactivation, the rate of increase is modest and variable[8,9].

Growing evidence suggests that rapid field cancerization can occur in the intestine as a result of changes to the crypt microenvironment, epithelial injury, and age. First, perturbations to the microenvironment can lead to the selective loss of WT ISCs and their rapid replacement by more fit premalignant ISCs. The increase in ISC replacement results in the accelerated fixation of somatic mutations within intestinal crypts and the efficient initiation of a field cancer[12]. Second, chronic epithelial injury induces crypt fission and can spread field cancers throughout the entire colonic epithelium in less than 4 years[4,13]. Third, rapid field cancerization can also occur if somatic mutations are acquired during intestinal development when more than 20% of the crypts are actively undergoing crypt fission[14,15]. However, somatic mutations that overcome the constraints of intestinal homeostasis and drive rapid field cancerization in otherwise healthy adult intestine have still not been found.

Rspondin-3 (RSPO3) is a microenvironmental factor that regulates ISCs and when overexpressed induces crypt hyperplasia[16]. Oncogenic fusions of RSPO3 with the protein tyrosine phosphatase receptor type K (PTPRK) are also known drivers of colon cancer[17,18]. RSPO3 and its oncogenic fusions are compelling candidates that could drive the rapid spread of intestinal field cancers. Current mouse models lack the resolution to easily investigate the cellular and molecular roles of RSPO3 in field cancerization. Convenient solutions also do not exist for expressing and directly comparing multiple RSPO3 mutations within a single isogenic mouse. Coincidentally, mouse models for broadly investigating the functional genomics of field cancerization are also needed. Therefore, we have developed a cancer rainbow (Crainbow) mouse modelling platform that combines

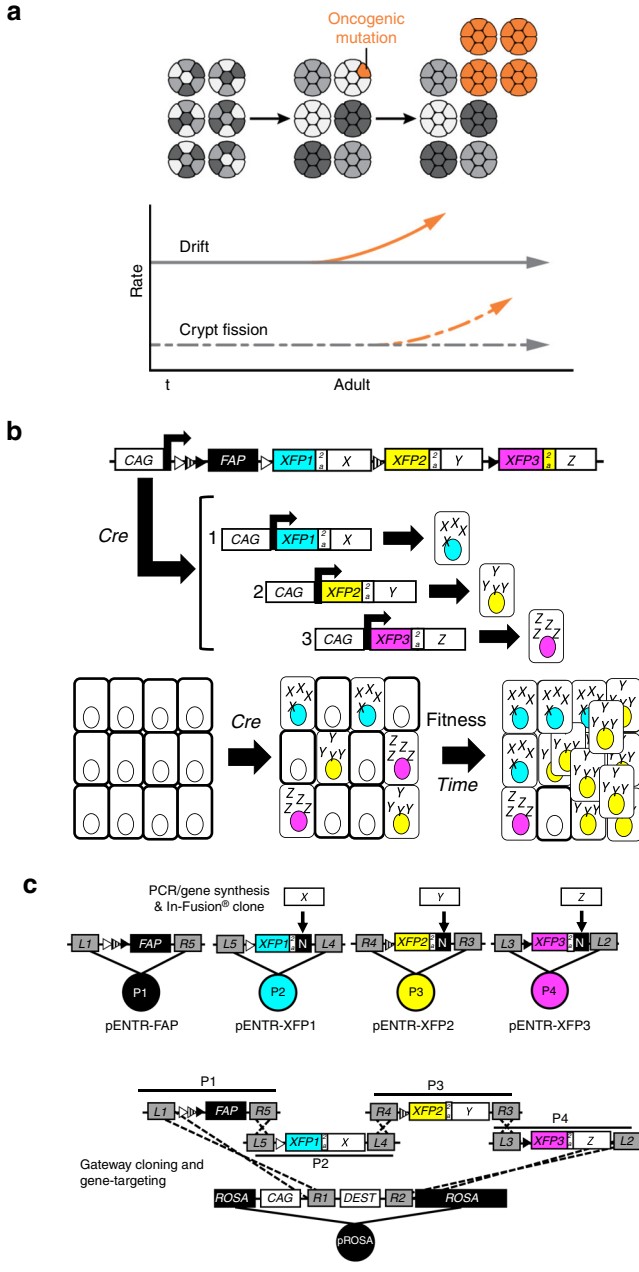

**Fig. 1** Engineering cancer rainbow mice. **a** The current two-step model of intestinal field cancerization. Somatic mutations increase stem cell fitness and lead to the fixation of premalignant clones within crypts (orange). Crypt fission results in duplication of premalignant crypts and the lateral propagation of somatic fields. **b** The Crainbow expression vector incorporates 4-tandem cassettes downstream of the ubiquitous chicken-β-actin promoter (CAG). The construct confers ubiquitous expression of a membrane-targeted and chemically inducible near-infrared fluorogen-activating peptide (FAP-Mars1) as a control. Cell type specific activation of Cre recombinase mediates recombination at one of three positions through inclusion of three pairs of orthogonal lox sites (LoxN: white triangle, Lox2272: hatched triangle, and LoxP: filled triangle). Single-copy transgene insertion provides a single outcome per cell and barcoding of each genetic fate by fluorescent imaging. **c** Candidate tumor driver genes are synthesized/PCR amplified and then In-Fusion® cloned into a bicistronic expression cassette. Multi-site Gateway™ cloning is used to directionally clone each pENTR plasmid into a Gateway™-compatible ROSA-targeting vector in a single step[21].

the desirable features of "Brainbow"[19,20] based lineage tracing with functional genomics screening into one seamless and interchangeable platform. Crainbow provides a means to induce multiple somatic mutations and visualize two essential attributes of field cancerization—ISC competition and clone spreading.

Crainbow modeling directly demonstrates that somatic mutations in the neonatal intestine clonally spread throughout the intestine during a critical period of intestinal growth and development[15]. In addition, *RSPO3* and its fusion isoforms are identified as a class of oncogenes that extrinsically transforms ISC behavior resulting in the widespread expansion of oncogenes throughout the adult epithelium in only a few weeks. Crainbow modelling is a transformative modelling technology and is a broadly applicable tool for visualizing the cellular and molecular dynamics of the early events that drive cancer.

## Results

**Engineering and validating cancer rainbow mouse models.**
Crainbow is a genetic model system for labelling and visualizing individual cells that express somatic mutations. Included in the Crainbow transgene are four positions that either express an inert fluorescent protein (position 0) or three spectrally resolvable fluorescent proteins paired with an oncogenic mutation of choice (positions 1–3). In addition, these candidate driver genes are fused to unique epitopes to ensure that their resultant protein products can be immunolocalized in tissue. In this manner, simple activation by Cre recombinase can induce spatiotemporal expression of fluorescently barcoded tumor driver genes and single-cell visualization of cell fitness, cell signalling, and the clonal spread of oncogenic mutations (Fig. 1b). In this report, several adaptations were made to overcome previous limitations in construct engineering and imaging[21]. First, a seamless and efficient cloning approach for building Crainbow targeting vectors was utilized[21–24] (Fig. 1c). Second, the fluorescent protein (XFP) palette for imaging in vivo and ex vivo was optimized. This optimization included the use of a chemically inducible near-

infrared fluorescent protein (FAP-Mars1[21]) for mitigating an overabundant control signal[20], targeting XFPs to the nucleus for cell counting, and delivery of two separate vector systems with different XFP palettes (Supplementary Figs. 1–3). Third, transgene expression was increased by including the woodchuck hepatitis posttranscriptional regulatory element (WPRE)[25]. Fourth, positional bias inherent to Cre/LoxP systems was empirically determined in a control mouse. Three Crainbow mouse models are developed in this study and are summarized in Table 1.

A Crainbow control mouse was developed to validate transgene expression and calculate positional bias. The *NCAT Crainbow* control mouse model (N-terminally truncated allele of ßcatenin) encodes the same N-terminally truncated and oncogenic form of ßcatenin (*ΔNßcat*) at each position[26]. Each *ΔNßcat* is coexpressed with *TagBFP, mTFP1, or mKO* (Fig. 2a, Supplementary Fig. 4). Simple confocal imaging for each fluorescent protein lineage marker can be used to assess recombination induced positional bias and the clonal spread of oncogenic mutations. *NCAT* Crainbow mice were bred to mice that express *Villin-Cre* recombinase to generate *NCAT(+/−):VilCre(+/−)* mice (*NCAT-VilCre*) and thereby induce recombination of Crainbow in the developing intestinal epithelium (e12.5)[27]. *NCAT^VilCre* mice were sacrificed between postnatal days (PND) 9–20 and the intestine was imaged by confocal microscopy. Since TagBFP quenched after fixation, imaging was performed on freshly isolated tissue when possible. Wholemount preparations of *NCAT^VilCre* intestine allowed confocal imaging through the serosa and into the intestinal crypt (Fig. 2b). Intestinal crypts harbor a population of ISCs that symmetrically divide and stochastically differentiate. This results in neutral drift behavior of the ISC and crypts that are populated by a single clone of stem cells—a process also known as crypt fixation[28,29]. Confocal imaging of intestinal crypts revealed that crypt fixation had already occurred in most crypts by the time of analysis, as evidenced by monoclonal crypts that expressed TagBFP (cyan), mTFP1 (yellow), or mKO (magenta)

---

**Table 1 Overview of cancer rainbow (Crainbow) mouse lines.**

| Symbol | Name | Oncogene[a] | Key observations | Map |
|---|---|---|---|---|
| NCAT | N-terminal truncated ßcatenin | Cre Off: Mars1 FAP  Cre On: 1. ΔNßcat 2. ΔNßcat 3. ΔNßcat | The positional recombination efficiency of Crainbow is biased 2.3-fold at position 1, 0.36-fold at position 2, and 0.32-fold at position 3. | Supplementary Fig. 4, Supplementary Data 1 |
| MCAT | Multiple isoforms of ßcatenin | Cre Off: Mars1 FAP  Cre On: 1. ΔNßcat 2. Ccat/Lef1 3. ΔNßcatΔC | Clone expansion occurs rapidly during perinatal development but is constrained by intestinal epithelial homeostasis in adult. | Supplementary Fig. 5, Supplementary Data 2 |
| ROBO | RSPO3 Crainbow | Cre Off: Mars1 FAP  Cre On: 1. RSPO3 2. PTPRK^e1:RSPO3^e2−5 3. PTPRK^e1−7:RSPO3^e2−5 | *RSPO3* oncogenes expand the crypt microenvironment to induce crypt fission and promote rapid oncogenic clone spreading in the adult epithelium. | Supplementary Fig. 7, Supplementary Data 3 |

[a]Description of oncogenes and Crainbow lines. (*NCAT*) *ΔNßcat* (aa. 80–781) is based upon the previously described N-terminal truncation mutant and escapes degradation to increase Wnt-signaling[26]. NCAT mice express *ΔNßcat* in positions 1–3 and are used to calculate positional bias in Crainbow transgenes. (*MCAT*) MCAT mice express three isoforms of *ßcat* that act as oncogene prototypes. *ΔNßcat* was directly compared with ßcat isoforms that possess an increased Wnt-signaling potential (*Ccat/Lef1: aa.693-781/aa.57-398*)[33,34] or a decreased Wnt-signaling potential (*ΔNßcatΔC, aa.80-693*)[33–35]. *MCAT* mice are used to test the comparative ability of Crainbow modelling and measure oncogenic clone competition of three oncogenes. (*ROBO*) RSPO3 is expressed by the crypt microenvironment and controls stem cell homeostasis[50]. The recently described oncogenic fusions of *PTPRK^e1:RSPO3^e2−5* and *PTPRK^e1−7:RSPO3^e2−5* are known drivers of colorectal cancer[16–18]. ROBO mice are used to assess how oncogenic activation of microenvironmental cues could also be potent drivers of the rapid spread of premalignant clones in the adult intestinal epithelium. Oncogenes are pseudocolored to match the fluorescent protein barcode

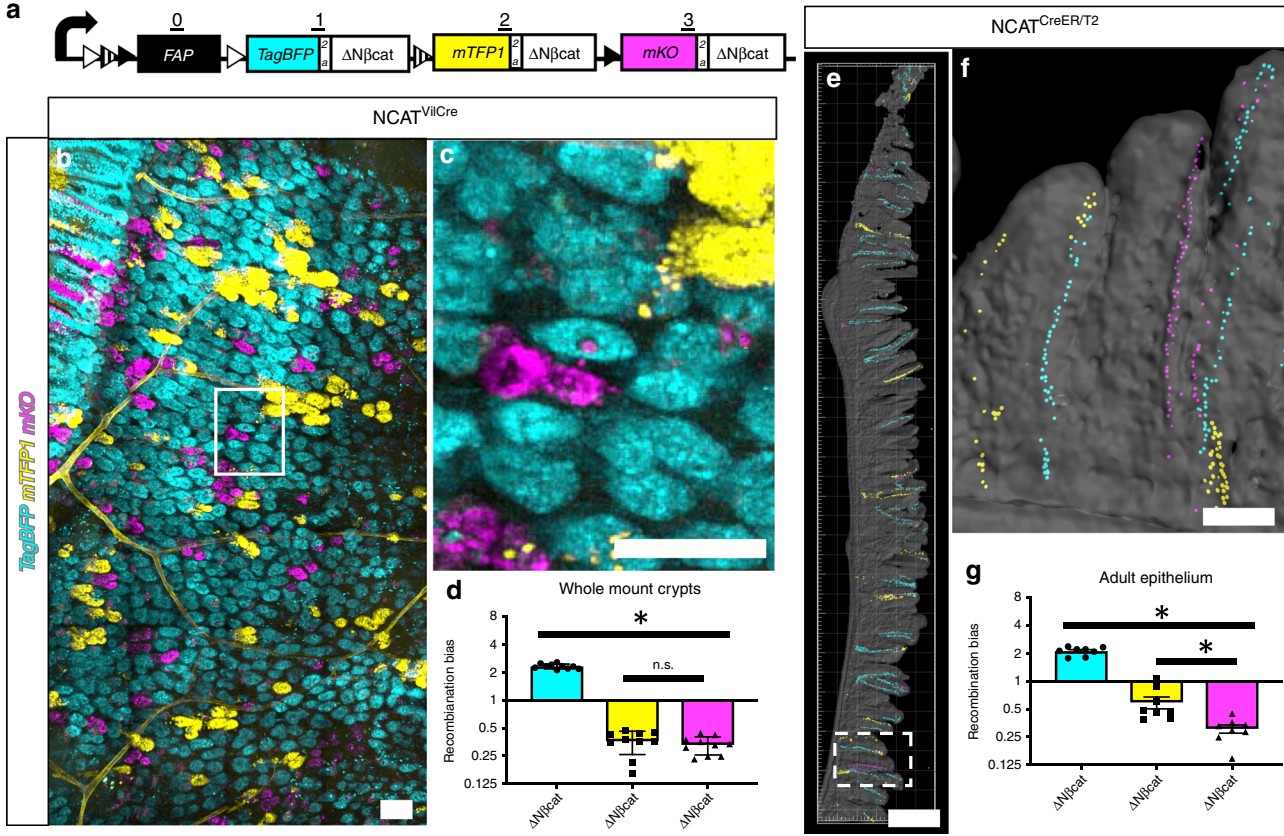

**Fig. 2** Validating Crainbow functionality. **a** Diagram of *NCAT-Crainbow* mice. See also Table 1 and Supplementary Fig. 1. **b** *NCAT*$^{VilCre}$ small intestine ($N = 10$ mice, PND9–PND20) was prepared as a whole-mount and confocal imaged for XFP expression in the intestinal crypts. **c** Inset in "**b**" at higher magnification. **d** Crypts were color segmented and counted. The experimentally observed value was normalized to the predicted stochastic outcome (0.33/XFP). The asterisk denotes statistical significance by one-way ANOVA (TagBFP vs. mTFP1: $p < 1e-6$, TagBFP vs. mKO: $p < 1e-6$, mTFP1 vs. mKO: not significant). NCAT$^{CreER/T2}$ mice ($N = 8$, 19–22 weeks of age) were injected with 200 mg/kg tamoxifen. Mice were sacrificed 12 days post tamoxifen injection, and the small intestine was vibratome sectioned and antibody stained to recover the quenched TagBFP signal. **e** Sections were imaged by confocal microscopy for XFP expression (TagBFP: cyan, EYFP: mTFP1, mKO: magenta; segmented nuclear masks shown and overlaid with surface rendered tissue outline). **f** Rotated inset in "**e**" at higher power. **g** Nuclei were segmented, counted, and normalized to the predicted stochastic outcome (0.33/XFP). The asterisk denotes statistical significance by one-way ANOVA (TagBFP vs. mTFP1: $p < 1e-6$, TagBFP vs. mKO: $p < 1e-6$, mTFP1 vs. mKO: 8.4e-4). (SEM included for each graph). Scale Bars = 100 μm in **b, c, f** and 2 mm in **e**. Source data are provided as a "Source Data file".

(Fig. 2b, c). Unbiased recombination of Crainbow should result in a 1/3 chance of activating positions 1, 2, or 3. Therefore, XFP crypts were counted and normalized to this expected probability of 0.33 to calculate recombination bias. Values equal to 1 reflect no bias, whereas >1 are positively biased and <1 are negatively biased. Crainbow activation has a 2.3-fold positive bias for position 1 and a concomitant negative bias that was similar for positions 2 and 3 (Fig. 2d). *NCAT* mice were also bred to *ROSA-Cre*$^{ER/T2}$ mice to generate *NCAT*(+/−):*ROSACre*$^{ER/T2}$(+/−) mice (*NCAT*$^{CreER/T2}$) and treated with tamoxifen to assess positional bias in adult mice. *NCAT*$^{CreER/T2}$ were sacrificed 12 days post tamoxifen injection to allow for ISC labelling and clonal replacement of the differentiated cell lineage. As expected, stripes of TagBFP, mTFP1, and mKO recombined epithelial cells emanating from the crypt base were observed by confocal microscopy of vibratome sliced small intestine (Fig. 2e, f). Quantification of recombination bias revealed a positive bias for position 1 and negative bias for positions 2–3. This bias was similar to the bias observed in developmental *NCAT*$^{VilCre}$ studies (Fig. 2g). The *NCAT* studies provided a basis for visualizing the clonal spreading of oncogenes in the intestine using Crainbow mouse models.

**Intestinal growth accelerates the clonal spread of oncogenes.** Genetic inactivation of adenomatous polyposis coli (*APC*) potentiates Wnt signalling and is a known driver of colon cancer[30–32]. ßcatenin is a downstream signaling effector and a prime candidate for increasing stem cell fitness and initiating field cancers. Several isoforms of ßcatenin have been used previously to study Wnt-signaling. *ΔNßcat* is already described above in *NCAT* mice and potentiates Wnt-signaling by escaping degradation. The *Ccat/Lef1* isoform is a fusion of the C-terminal transactivation domain of ßcat to the transcription factor Lef1. *Ccat/Lef1* potentiates Wnt signalling but cannot be sequestered by epithelial cadherin (CDH1)[33,34]. *ΔNßcatΔC* lacks the N-terminal domain of ßcat and escapes degradation but also lacks the C-terminal transactivation domain and is thereby unable to transduce nuclear Wnt signalling[33–35]. These somatic mutations vary in their Wnt-signaling aptitude and are excellent prototypes for validating Crainbow as a robust system that quantifies the clonal expansion of oncogenes in vivo. Therefore, an *MCAT-Crainbow* mouse model (Multiple forms of ßcatenin, Fig. 3a and Supplementary Figs. 3, 5) was engineered. MCAT also used *mTFP1, EYFP*, and *mKO* palette for more favorable confocal imaging.

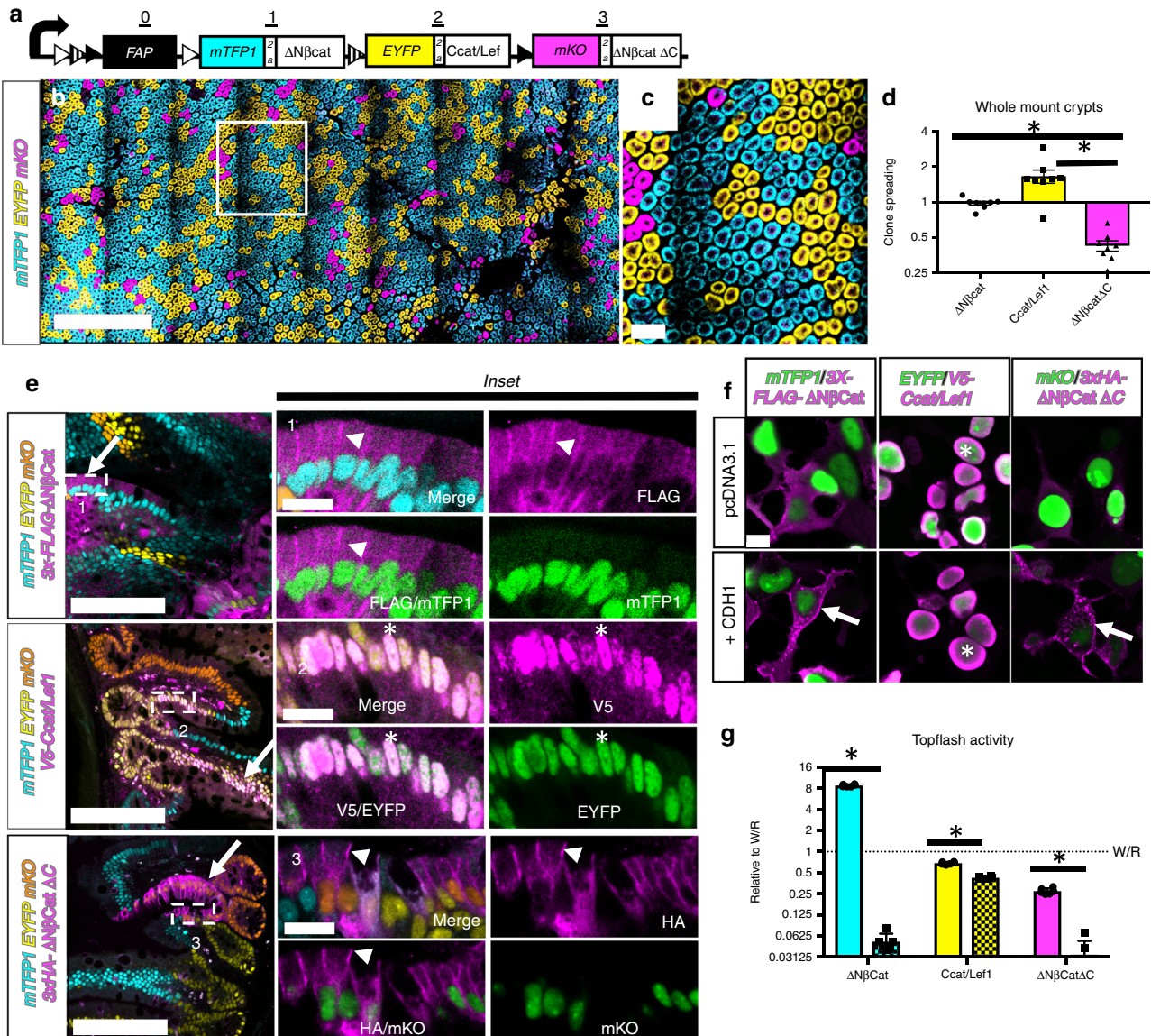

**Fig. 3** Widespread expansion of oncogenic clones during perinatal development. **a** Diagram of *MCAT-Crainbow* mice. See also Table 1 and Supplementary Fig. 5. **b** *MCAT^VilCre* small intestine (*N* = 10 mice, 3–6 weeks of age) prepared as a wholemount and confocal imaged. **c** Inset in "**b**" at higher magnification. **d** *MCAT^VilCre* Crypts were color segmented, counted and normalized to the positional bias calculated in *NCAT^VilCre* mice. Asterisk denotes statistical significance by one-way ANOVA (mTFP1 vs. EYFP: *p* = 0.003, mTFP1 vs. mKO: *p* = 0.016, EYFP vs. mKO = 3e-6). **e** Immunostaining for FLAG, V5, or HA epitopes (magenta) specific to each ßcat isoform in MCAT^VilCre small intestine vibratome slices and merged with fluorescent lineage markers (mTFP1: cyan, EYFP: yellow, and mKO: orange). Arrows denote isoform expression with cognate lineage reporter (FLAG and mTFP1, V5 and EYFP, and HA and mKO). Corresponding insets depict higher magnification images. Arrowheads denote membrane-localized ßcat, whereas asterisk denotes nuclear-localized ßcat. Epitope stains (magenta) are also presented as merged and as a single-channel image with its cognate fluorescent lineage reporter (green). **f** HEK cells were transiently transfected with MCAT isoforms, fixed, stained, and imaged for the indicated epitope (magenta) and fluorescent reporter (green). Cells were also cotransfected with epithelial cadherin (*CDH1*) as indicated. Arrows denote sequestration of ßcat at the plasma membrane, and the asterisk denotes nuclear ßcat. **g** Wnt signalling activity for each oncogene in the absence of *CDH1* (solid bar) or in the presence of overexpressed *CDH1* (hatched bar) (*N* = 6 wells per condition and independently repeated in four experiments). TOP FLASH activity was normalized to WNT/RSPO-stimulated control cells (dashed line). Asterisk denotes statistical significance by two-way ANOVA and Bonferroni's multiple comparisons test (cyan < 1e-6, yellow = 0.01, magenta = 0.02). (SEM included for each graph). Scale Bars = 1 mm in **b**, 100 μm in **c/e**, 15 μm in **e:** insets 1–3, and 10 μm in **f**. Source data are provided as a "Source Data file".

MCAT Crainbow mice were bred to mice that express *Villin-Cre* recombinase to generate *MCAT*(+/−):*VilCre*(+/−) mice (*MCAT^VilCre*) and thereby induce recombination of Crainbow in the developing intestinal epithelium (*e12.5*)[27]. Whole-mount confocal imaging of 3–6-week-old *MCAT^VilCre* small intestine revealed that crypt fixation had already occurred, as was evident by a single color of epithelial cells in each crypt (Fig. 3b, c). The

spread of oncogenic clones through the intestine was also studied in MCAT mice. The same *ΔNßcat* allele was used in Position 1 of *NCAT* and *MCAT* mice. Therefore, a relative clone spreading potential was calculated by normalizing the MCAT crypt frequency to the positional recombination bias calculated in *NCAT^VilCre* mice. *Ccat/Lef1* expression induced a relative 1.6-fold increase in clone spreading potential. In contrast, *ΔNßCatΔC*

induced a respective 0.4-fold reduction in clone spreading potential and ΔNßcat was unchanged (Fig. 3d). The proximal to distal axis of the intestine differs in Wnt/ßcatenin signalling potential[36]. ΔNßCatΔC expression resulted in a similar decrease in clone spreading potential in the colon and the effects of Ccat/Lef1 were mitigated and more similar to ΔNßcat (Supplementary Fig. 6a, b).

Epithelial cadherin (CDH1) sequesters ßcat at the lateral plasma membrane acting as a molecular "sink" for excess ßcat that ultimately inhibits Wnt-signalling[37,38]. We hypothesized that the increased spread of Ccat/Lef1 clones was due to the loss of CDH1 sequestration activity. Antibody staining for the epitope tags fused to each ßcat isoform revealed that each isoform was correctly expressed with its cognate XFP (Fig. 3e, arrows). FLAG immunostaining for ΔNßcat and HA immunostaining for ΔNßCatΔC showed plasma membrane sequestration of ßcat with no detectable nuclear signal (Fig. 3e, arrowheads). In contrast, V5 immunostaining for Ccat/Lef1 revealed nuclear localization with no detectable expression at the plasma membrane (Fig. 3e, asterisks). A similar effect was observed in a complementary in vitro transfection system (Fig. 3f). In addition, a TOP-FLASH luminescent assay for Wnt signalling[39] demonstrated that only Ccat/Lef1 could effectively transduce Wnt/ßcat signalling in the presence of CDH1 (Fig. 3g). These results confirmed that Crainbow modelling could be used for inducing and directly comparing multiple premalignant epithelial clones harboring oncogenic mutations and that these clones effectively spread during perinatal intestinal development.

*MCAT* mice were also used to study the evolution of oncogenic clones in adult intestine. *MCAT* mice were bred to *ROSA-Cre*[ER/T2] mice to generate *MCAT(+/−):ROSACre*[ER/T2](+/−) mice (*MCAT*[CreER/T2]). Adult *MCAT*[CreER/T2] mice were treated with 200 mg/kg tamoxifen and chased for 3 days or 8 weeks. Sparse recombination was observed throughout the villus and crypt epithelium at day 3 (Fig. 4a). Prior lineage tracing studies of the intestinal epithelium revealed that recombined villus epithelial cells are lost within a few days and are replaced by clones of epithelial cells emanating from recombined ISCs[29]. Clones of MCAT recombined cells could be similarly observed 8 weeks after tamoxifen induced recombination (Fig. 4b). Clones were restricted to narrow stripes of recombined epithelial cells within each villus and were typically only a few cells wide. The widespread recombination of an entire villus was not observed. An estimate for clone spread was determined by counting the total recombined cells at day 3 versus 8 weeks. The total number of recombined cells did not increase from day 3 to 8 weeks for any of the oncogenic ßcat isoforms. An apparent but statistically insignificant decrease in total recombined Ccat/Lef1 cells was also observed (Fig. 4c). Therefore, clone spreading does not significantly occur in adult *MCAT* intestine.

The fixation of somatic mutations within a crypt is an initiating event in cancer field formation. A previous report documented that ISCs with potent somatic mutations can still be replaced by neighboring stem cells within the crypt[10]. Therefore, crypt fixation rate was measured in MCAT mice. *MCAT*[CreER/T2] mice were treated three times with 200 mg/kg tamoxifen (Days 1, 3, and 5) to abundantly label multiple ISCs within the crypts. Mice were sacrificed at Day 7 or at 8 weeks. Whole-mount confocal imaging of intestinal crypts at day 7 revealed that 42% of crypts were labelled. Multiple examples of monoclonal, biclonal, or triclonal crypts were evident (Fig. 4d). Prior lineage tracing studies demonstrated that crypt fixation occurs within 4–8 weeks[29]. Whole-mount confocal imaging of MCAT intestinal crypts 8 weeks after recombination revealed that crypt fixation had occurred in 15% of the crypts counted (Fig. 4e). The relative fraction of fixed crypts for each MCAT lineage was calculated at

8 weeks and normalized for positional bias in order to calculate the relative crypt fixation potential of each ßcat isoform. The results indicated that the relative fixation potential of mTFP1:ΔNßcat ISCs was modestly increased compared to the reduced fixation potential of EYFP:Ccat/Lef1 and mKO:ΔNßCatΔC ISCs (Fig. 4f). The direct measurement of clone spreading and crypt fixation within *MCAT* mice indicates that the normal adult intestine is resistant to the clonal expansion of oncogenic mutations.

Interestingly, efficient and widespread field cancerization during development has previously been attributed to the plasticity of crypts during development—including crypt fission and nascent crypt formation[14]. Intestinal organoid cultures effectively model crypt plasticity[40]. Therefore, MCAT organoids were used as an ex vivo correlate of the clonal expansion of oncogenes observed during development. ΔNßcat−, Ccat/Lef1−, ΔNßCatΔC−expressing crypts were isolated from *MCAT*[VilCre] mice and similarly established viable organoids in full media (Fig. 4g, h, Panel 1). MCAT organoids were passaged so that each lineage could be longitudinally studied for alterations in competitive fitness over time. Interestingly, *CCat/Lef1* expressing organoids rapidly expanded after multiple rounds of subculturing, whereas *ΔNßcat* and *ΔNßCatΔC* expressing organoids did not (Fig. 4i and Supplementary Fig. 6c). Paneth cells are the source of Wnts in organoid culture and are required for organoid growth[41]. Organoids were treated with c59 to prevent Wnt acylation and thereby inhibit Wnt secretion[42]. Only Ccat/Lef1 expressing organoids grew in the absence of Wnt (Fig. 4g, h, j, Panel 2). Similarly, organoids require exogenous Rspondins (RSPO)s for their growth[43]. Only Ccat/Lef1 organoids could grow without Rspondin (Fig. 4g, h, j, Panel 3). Ccat/Lef1 organoids could also grow in the absence of Wnts and RSPO (Fig. 4g, h, j, Panel 4). Therefore, the oncogenic activation of Wnt/ßcat signaling endows ISCs with increased fitness that results in their preferential expansion during a permissive period of intestinal development but not in healthy adult intestine.

## Rapid spreading of oncogenes in the adult intestine by microenvironmental oncogenesis. The abundant crypt fission and formation of nascent crypts in organoids is likely due in part to the artificial reconstitution of the microenvironment, in which the cells are grown, including important niche factors like (RSPO)s[41,43]. *RSPO3* is a critical regulator of the intestinal micro-environment[16] and in vivo its expression is normally limited to the subepithelial mesenchyme of *WT* intestine (Fig. 5a). *RSPO3* fusions with *PTPRK* have been found in colon cancer patients[17] and are protumorigenic in mice[18]. Therefore, a Crainbow *RSPO3* model was generated to test the hypothesis that misexpression of *RSPO3* and its fusion products perturb the microenvironment of the crypt and induce the spread of oncogenic clones by increasing crypt fission. *ROBO* Crainbow mice (RSPO3-Crainbow) encode *RSPO3*, *PTPRK*[e1]:*RSPO3*[e2−5], and *PTPRK*[e1−7]:*RSPO3*[e2−5] at positions 1–3 of Crainbow and are coexpressed with the *mTFP1, EYFP*, and *mKO* fluorescent protein barcode (Fig. 5b and Supplementary Fig. 7).

*ROBO* Crainbow mice were bred to mice that express *Villin-Cre* recombinase to generate *ROBO(+/−):VilCre(+/−)* mice (*ROBO*[VilCre]) and thereby induce recombination of Crainbow in the developing intestinal epithelium (*e12.5*)[27]. The resultant *ROBO*[VilCre] pups developed distended abdomens and were moribund by the time of weaning. Gross anatomical examination revealed dilated and elongated small and large intestines compared with WT mice (Fig. 5c). Whole-slide imaging of H&E stained Swiss-roll *WT* and *ROBO*[VilCre] small and large intestines revealed an expansion of the crypt zone in the small

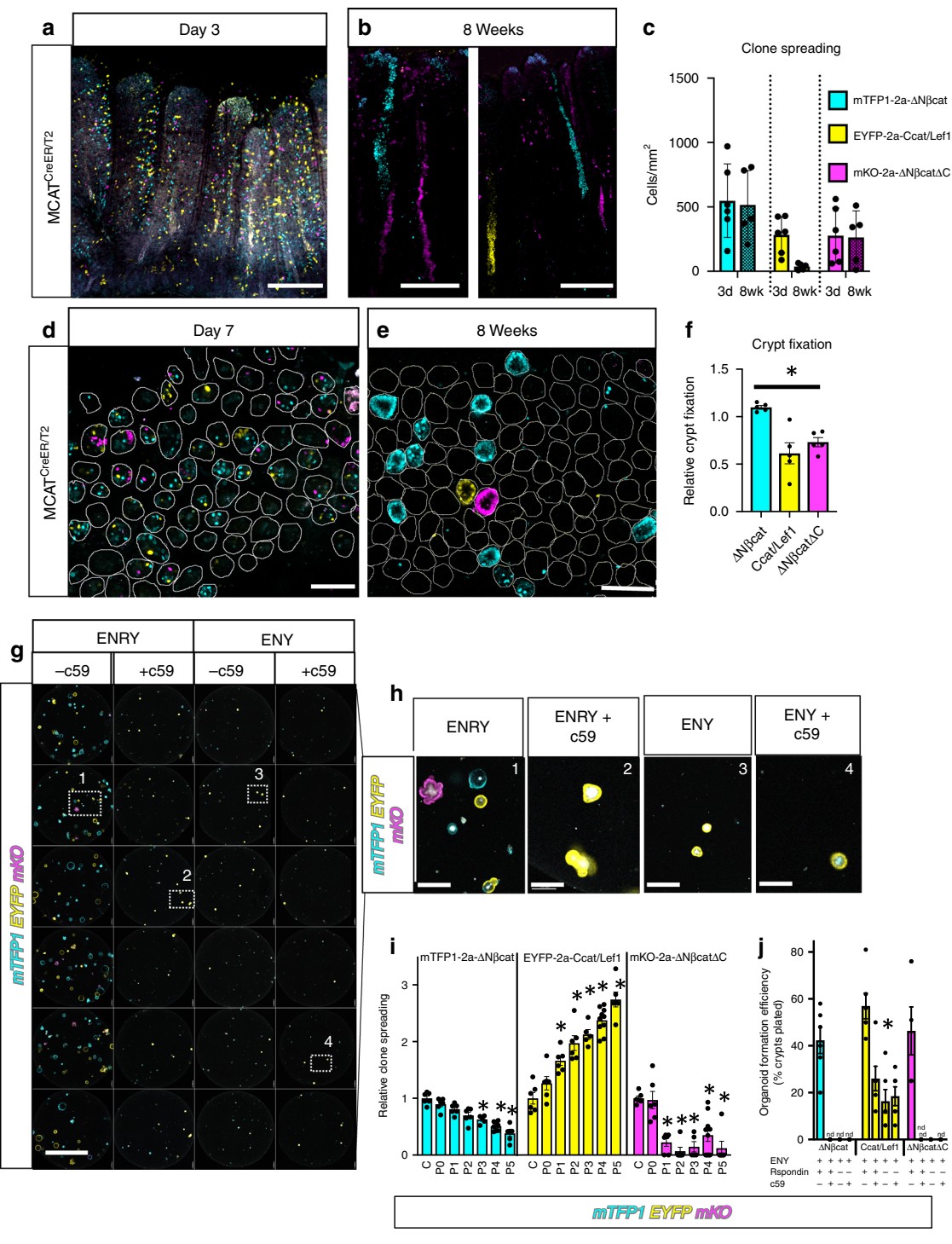

intestine, basal crypt hyperplasia, architectural distortion, and occasional luminal crypt abscesses (Fig. 5d, Supplementary Figs. 8–13). Similarly, the colonic crypts were expanded by basal crypt hyperplasia and architectural distortion including crypt branching and budding (Fig. 5e, Supplementary Figs. 8–13). RNA FISH was used to molecularly map the stem cell compartment in *WT* and *ROBO^VilCre* intestine. A potent expansion of the stem cell compartment in *ROBO^VilCre* mice was observed and included an overabundance of *Lgr5*-positve stem cells and *Top2a*-positive transiently amplifying (TA) cells[44] (Fig. 5d, e, Supplementary Figs. 8–13). Conversely, a marked reduction in differentiated *Alpi*-positive enterocytes[45] was

observed in *ROBO^VilCre* intestine. *Reg3b*, a recently described marker of enterocytes adjacent to the stem cell crypt[45,46], was found in scattered cells interposing the TA zone of the crypt and more mature enterocytes of the villus. *Reg3b* expressing cells were also expanded in *ROBO^VilCre* small intestine when compared with *WT*. In the colon, *Reg3b* expressing cells were less frequent in both *WT* and *ROBO^VilCre* mice (Fig. 5d, e, Supplementary Figs. 8–13). *ROBO^VilCre* intestine also had an expanded proliferative zone and was in contrast to the normal proliferative zone found in *MCAT^VilCre* intestinal crypts (Fig. 5f). *ROBO^VilCre* organoids also grew in the absence of exogenous RSPOs (Fig. 5g). These data confirm that ectopic overexpression

**Fig. 4** The crypt microenvironment impairs the spread of oncogenic clones in adults. **a, b** *MCAT*[CreER/T2] mice (>6 weeks of age) were I.P. injected with 200 mg/kg of tamoxifen and sacrificed 3 days (N = 8) or 8 weeks later (N = 5). The small intestine was vibratome sectioned and imaged. **c** Nuclei were segmented, counted, and normalized to the total volume of tissue imaged. Clone spread was determined by comparing the change in total recombined nuclei for each fate at day 3 and 8 weeks. Statistical significance by mixed effects modelling and Sidak multicomparison correction (ΔNßcat: p = 0.99, Ccat/Lef: p = 0.14 ΔNßcatΔC: p = 0.99). **d, e** *MCAT*[CreER/T2] (>6 weeks of age) were injected three times (Day 1, 3, 5) with 200 mg/kg tamoxifen and sacrificed at Day 7 (N = 4) or at 8 weeks (N = 5). The small intestine was whole mount imaged by confocal microscopy. Representative crypts are outlined and **f** the relative fraction of crypt fixation events for each ßcat isoform was measured. Statistical significance by one-way ANOVA and Holm-Sidak's multiple comparison test (ΔNßcat vs. Ccat/Lef1: p = 0.011 (ΔNßcat vs. ΔNßcatΔC: p = 0.0062, Ccat/Lef1 vs. ΔNßcatΔC p = 0.259). **g** *MCAT*[VilCre] crypts were isolated and cultured as in the indicated media (+/−c59 inhibitor at 10 nM) and then imaged by confocal microscopy. **h** High-magnification view of wells (N = 6) from each condition in "**g**" and rotated corresponding insets. **i** *MCAT*[VilCre] organoids were subcultured (P0–P5, and N = 5–10 wells per subculture) and the entire well imaged by confocal microscopy and organoids of each lineage were counted. Relative clone spreading for each organoid fate was calculated by normalization to the crypt number (C) at isolation (see also Supplementary Fig. 6c). The asterisk denotes statistical significance by mixed effects modelling and Dunnett post hoc relative to Crypt control (C) (ΔNßcat: p = 0.03, p = 0.002, p < 0.0001 and Ccat/Lef1: p < 0.0001 for each asterisk and ΔNßcatΔC p < 0.0001 for each asterisk). **j** Organoid formation efficiency for MCAT crypts cultured in "**g–h**." The asterisk denotes statistical significance by two-way ANOVA and Tukey's multiple comparisons test. ENRY conditions: not significant between ßcat isoforms, whereas Ccat/Lef1 was significantly different from ΔNßcat and ΔNßcatΔC for each treatment (ENRY + c59: p = 0.0003, ENY: p = 0.03, and ENY + c59: p = 0.01). (SEM included for each graph). Scale bars = 200 μm in **a**, **b**, 100 μm in **d**, **e**, 2 mm in **g**, and 200 μm in **h**. Source data are provided as a "Source Data file".

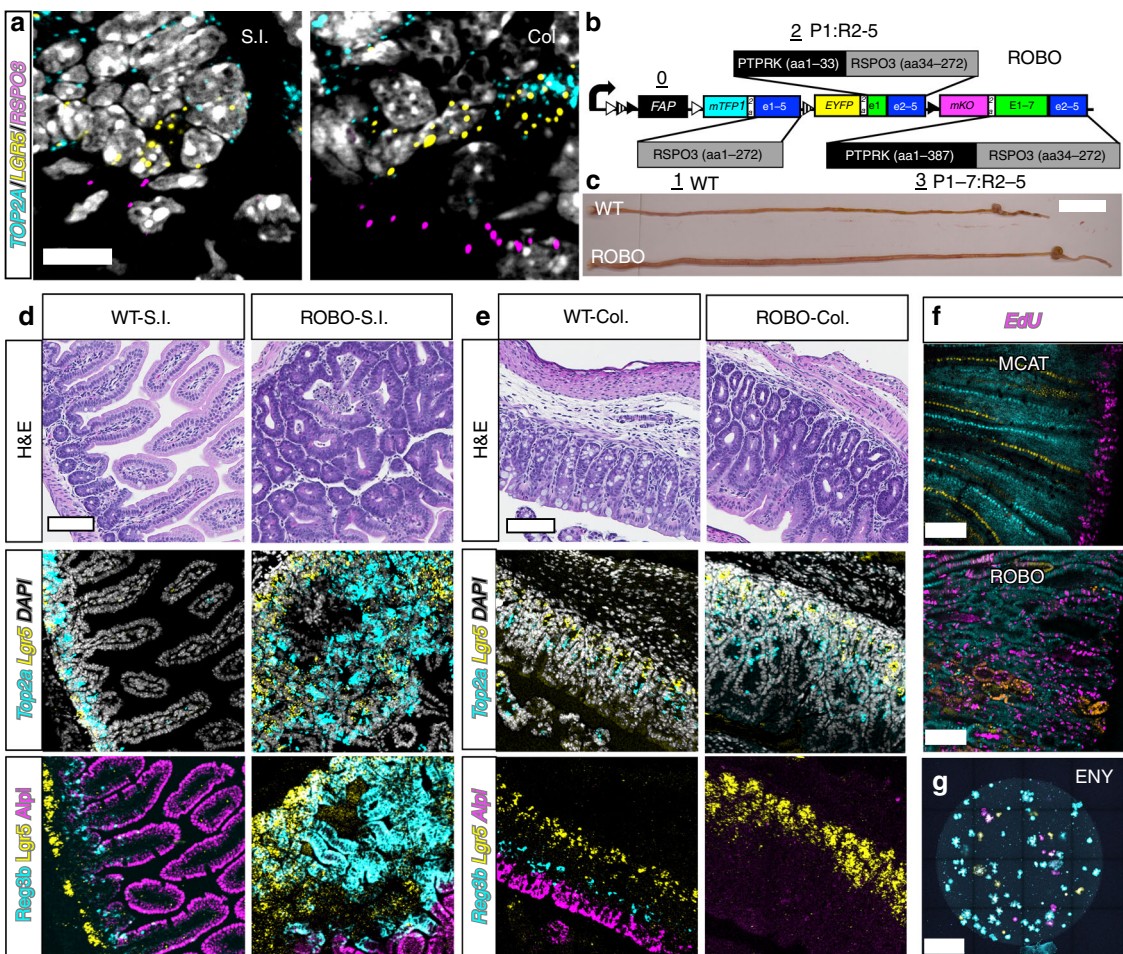

**Fig. 5** Ectopic expansion of the intestinal stem cell compartment by expression of oncogenic RSPO3. **a** Multiprobe RNA FISH and confocal microscopy for *Top2a* (cyan), *Lgr5* (yellow), and *Rspo3* (magenta) in paraffin-embedded sections of the small intestine (SI) and colon (Col). **b** *ROBO*[VilCre] mice expressing oncogenic human RSPO3. Position 0: control and membrane-targeted *FAP-Mars1*, Position 1: *mTFP1* (cyan) coexpressed with *3×FLAG-RSPO3*, *EYFP* (yellow) coexpressed with *V5-PTPRK*[e1]:*RSPO3*[e2–5], and Position 3: *mKO* (magenta) coexpressed with *3×HA-PTPRK*[e1–7]:*RSPO3*[e2–5] (see also Supplementary Fig. 7). **c** Gross examination of the *ROBO*[VilCre] mice and gastrointestinal tracts at PND16 compared with *WT* littermate controls. *WT* and *ROBO* **d** small intestines (SI) and **e** colons (Col) were paraffin embedded and sectioned for H&E pathology and coregistry of indicated cell type markers by multiprobe RNA FISH. Images are insets from whole-slide imaged Swiss rolls (see Supplementary Figs. 8–13). **f** *MCAT*[VilCre] and *ROBO*[VilCre] mice were injected with EdU (magenta) and coimaged for each lineage reporter by confocal microscopy (mTFP1: cyan, EYFP: yellow, mKO: orange). **g** *ROBO*[VilCre] organoids can grow without exogenous RSPO3 (ENY) and were imaged by confocal microscopy for each lineage reporter. Scale bars = 10 μm in **a**, 2 cm in **c**, 100 μm in **d, e**, 50 μm in **f**, and 1 mm in **g**. Source data are provided as a "Source Data file".

of Rspondins can restructure the microenvironment of the intestine leading to crypt hyperplasia and the ectopic expansion of proliferative stem cells throughout the epithelium[16,47]. We have termed this process microenvironmental oncogenesis.

Previous reports showed that RSPO3 and RSPO3:PTPRK fusions are protumorigenic in mice but the lack of lineage tracing prevented full understanding of how each form of the protein affects stem cell dynamics[18]. Therefore, we assessed stem cell clonality and lineage development in $ROBO^{VilCre}$ small intestine and compared this to $MCAT^{VilCre}$ small intestine. Confocal imaging revealed that $ROBO^{VilCre}$ intestine was significantly larger than age-matched $MCAT^{VilCre}$ intestine (Fig. 6a). At higher magnification, $MCAT^{VilCre}$ intestine displayed the typical pattern of single-color clones in the villus that emanate from the base of the intestinal crypt. In contrast, $ROBO^{VilCre}$ mice had fewer large single-color clones and instead were characterized by smaller clones, in which each lineage was heterogeneously mixed throughout the villus and crypt (Fig. 6b). This suggested that crypt fixation had been inhibited in $ROBO^{VilCre}$ mice. Whole-mount confocal imaging crypts revealed that the $ROBO^{VilCre}$ crypts were markedly polyclonal (Fig. 6c). This was in direct contrast to the monoclonality and crypt fixation evident in control $MCAT^{VilCre}$ crypts (Fig. 6d). Instances of monoclonality (Fig. 6e, insets 1–3), biclonality (Fig. 6e, insets 4–6), and triclonality (Fig. 6e, insets 7, 8) were observed in the $ROBO^{VilCre}$ crypts. In addition, crypt fission and crypt budding could be observed in $ROBO^{VilCre}$ crypts by confocal microscopy and confirmed the histopathology assessment (Fig. 6e, insets 1–3). Crypt fixation was scored in $ROBO^{VilCre}$ and compared with $MCAT^{VilCre}$ and the control $NCAT^{VilCre}$ mice. This analysis revealed that 95% and 97% of crypts were monoclonal in $MCAT^{VilCre}$ and $NCAT^{VilCre}$ mice, respectively. In contrast, only 76% of crypts were monoclonal or fixed in $ROBO^{VilCre}$ mice (Fig. 6f). Approximately 23% of crypts were biclonal in $ROBO^{VilCre}$ mice, whereas only 5% and 3% of crypts were biclonal in $MCAT^{VilCre}$ and $NCAT^{VilCre}$ mice, respectively. Triclonal crypts were found in 1% of the $ROBO^{VilCre}$ mice but were not observed in $MCAT^{VilCre}$ or $NCAT^{VilCre}$ mice. Crypt fixation was also inhibited in the colon (Fig. 6g, h). These data demonstrated that microenvironmental oncogenesis inhibits the neutral drift behavior of ISCs and attenuates the rate of crypt fixation.

Next, lineage dynamics and signalling were assessed by single-cell RNA sequencing (scRNAseq) of crypt enriched tissue isolated from WT and $ROBO^{VilCre}$ small intestine. t-SNE clustering and visualization of scRNAseq data revealed ten cell types. Two of these cell types were fibroblasts and immune cells. The remaining eight include stem cells, transiently amplifying cells, secretory Paneth and goblet cells, enteroendocrine cells, and three populations of enterocytes that could be qualitatively associated with distinct spatial zones within the villus[46] (Fig. 7a, Supplementary Fig. 14). Surprisingly, the overall fractional contribution of each cell type and the cell-cycle state did not change significantly between WT and ROBO crypts (Fig. 7b, c). Signalling was also largely unaffected as only 25 genes (23 genes in TA cells and two genes in ENTb1) were differentially expressed (Supplementary Data 4). The scRNAseq data suggest that microenvironmental oncogenesis extrinsically modifies the stem cell compartment.

Last, we hypothesized that microenvironmental oncogenesis can promote the expansion of oncogenic clones in adult intestine. To test this, ROBO mice were bred to ROSA-Cre$^{ER/T2}$ mice to generate $ROBO(+/-):ROSACre^{ER/T2}(+/-)$ mice. The resultant $ROBO^{CreER/T2}$ mice were injected with a single dose of tamoxifen and chased for either 3 days or 8 weeks. Both small and large intestinal mucosa showed crypts with dysplasia of the ISC compartment at 8 weeks (Supplementary Fig. 15). Confocal imaging of each lineage demonstrated that there was sporadic recombination throughout the crypt and villus epithelium at day 3 (Fig. 7d). We found that at 8 weeks recombined cells had spread throughout the $ROBO^{CreER/T2}$ intestinal epithelium and were therefore no longer constrained to a local clone within single crypts (Fig. 7e). This was in clear contrast to similarly performed experiments in $MCAT^{CreER/T2}$ mice (Fig. 4a–c). We quantified clone spreading by counting the total number of recombined cells at day 3 and 8 weeks. The results indicated oncogenic clones had significantly spread in $ROBO^{CreER/T2}$ mice. Specifically, WT RSPO3 clones increased by 6.7-fold, and $PTPRK^{e1}:RSPO3^{e2-5}$ clones increased by 1.55-fold whereas $PTPRK^{e1-7}:RSPO3^{2-5}$ clones decreased by 0.55-fold (Fig. 7f). In addition, the resultant crypts and clones were heterogenous, confirming that neutral drift and crypt fixation was also impaired in adults (Fig. 7g). Together, our data illustrate that microenvironmental oncogenesis can induce and rapidly spread heterogenous oncogenic clones in the adult intestine by increasing crypt fission, expanding the stem cell compartment, and inhibiting crypt fixation.

## Discussion

The current study developed a mouse platform useful for modeling the oncogenic and cellular determinants of ISC competition and the clonal spread of somatic mutations throughout the intestine. Crainbow mouse models were used to induce somatic activations and then visualize the evolution and competition of premalignant epithelial clones by fluorescent imaging. The spatiotemporal activation of somatic mutations was used to identify the time and microenvironment constraints present within the intestine that inhibit the clonal expansion of premalignant epithelium. Crainbow modelling revealed that the microenvironment inherently restricts this process to a susceptible period of intestinal development. However, in the adult, oncogenic activation of microenvironmental factors expands the stem cell compartment and leads to the rapid spread of oncogenic clones.

Crainbow lineage tracing experiments demonstrated that oncogenic activation of ßcat (MCAT-Crainbow) results in widespread expansion of premalignant clones during a critical period of perinatal development but not when activated in adult intestine (Fig. 8a). This illustrates how early APC inactivation can lead to increased tumor burden in adults[14]. The ISC compartment of mice is established during perinatal growth of the intestine and is initiated by the nascent formation of intestinal crypts[48]. The nascent crypts then multiply throughout the intestine by undergoing several rounds of crypt fission until a mature intestine is formed, at which time the rate of crypt fission slows[15]. MCAT Crainbow studies show that somatic activations can hijack the normal mechanics of intestinal growth and efficiently spread premalignant clones before adulthood. The growth of the human intestine occurs throughout the first 2 years of life and is also characterized by a sustained increase in crypt fission that markedly attenuates when the intestine reaches maturity[11]. Therefore, the infant's environmental burdens, diet, and injury should be closely monitored to guard against the spread of oncogenic clones. Future work should identify additional genetic markers that could be used as noninvasive measures of premalignant clones to identify children with increased risk of colon cancer in adulthood.

The lack of adenomas in ßcat Crainbow mice (NCAT or MCAT) was surprising—especially since Harada et al.[26] have shown previously that extensive intestinal polyps form when the N-terminal domain of endogenous ßcatenin is deleted. Our in vitro signaling assays confirmed ΔNßcat and Ccat/Lef1 isoforms were oncogenic as measured by Wnt/Rspondin

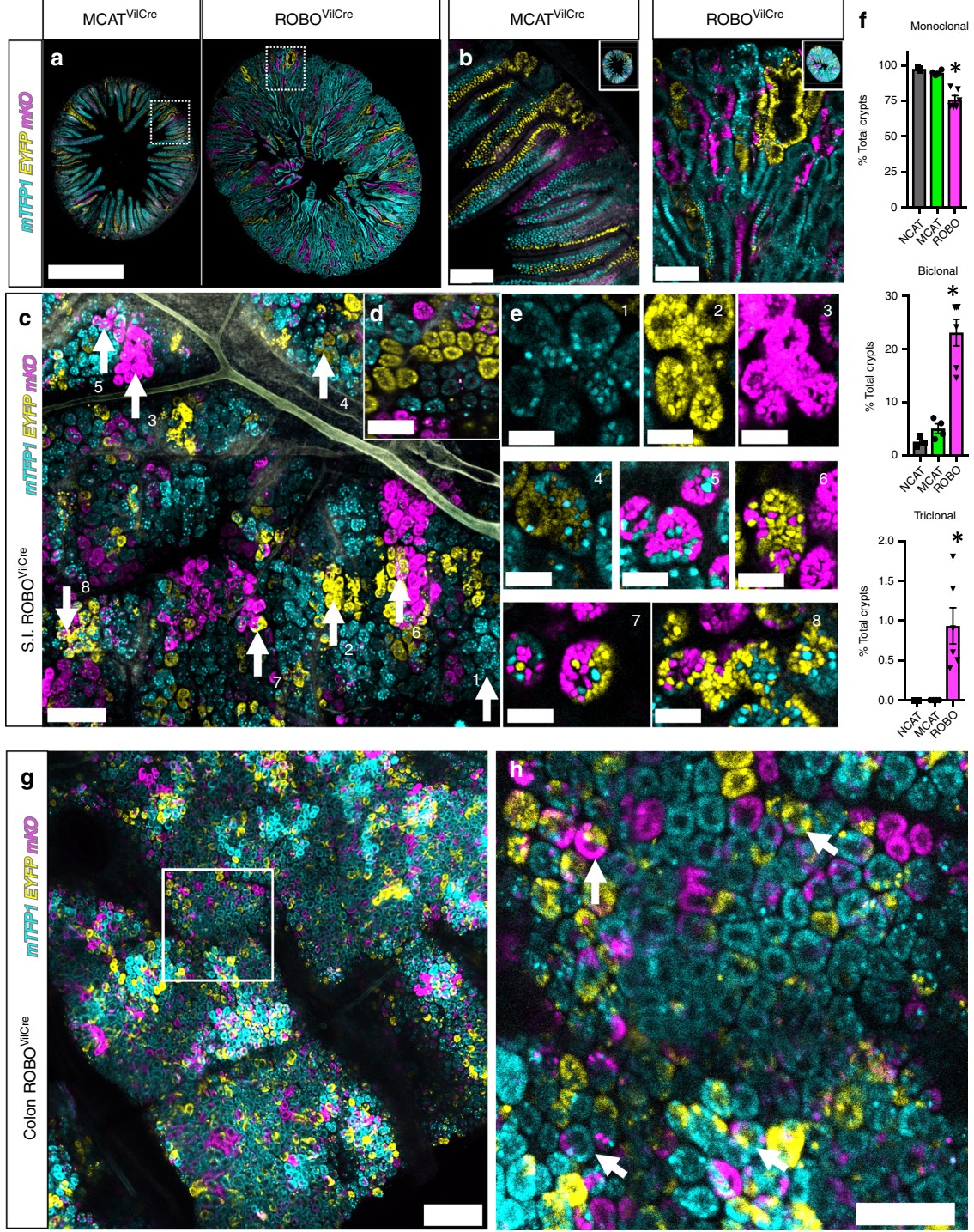

**Fig. 6** Microenvironmental oncogenesis attenuates crypt fixation. **a** Vibratome cross-sections of *MCAT^VilCre^* and *ROBO^VilCre^* small intestines were confocal imaged and tiled for each lineage reporter (PND17). **b** Higher magnification of areas indicated in "**a**". **c** Whole-mount confocal imaging of small intestines for each fluorescent lineage reporter in *ROBO^VilCre^* mice. **d** Whole-mount confocal imaging of *MCAT^VilCre^* small intestines. **e** Indicated callouts (1–8) from "**c**" at higher magnification. **f** The percentage of each crypts that are monoclonal (crypt fixation), biclonal, or triclonal was calculated for *NCAT^VilCre^* mice (*N* = 3 and 938 crypts), *MCAT^VilCre^* mice (*N* = 9, and 6006 crypts analyzed at PND17) and *ROBO^VilCre^* mice (*N* = 6 and 3557 crypts analyzed). (SEM included for each graph). **g** Wholemount confocal imaging of *ROBO^VilCre^* colon at PND18. **h** Region of interest in "**g**" at higher magnification. Arrows show examples of crypts where fixation has not occurred. Scale bars = 1 mm in **a**, 100 μm in **b/d, g, h**, 200 μm in **c**, and 40 μm in **e**. Source data are provided as a "Source Data file".

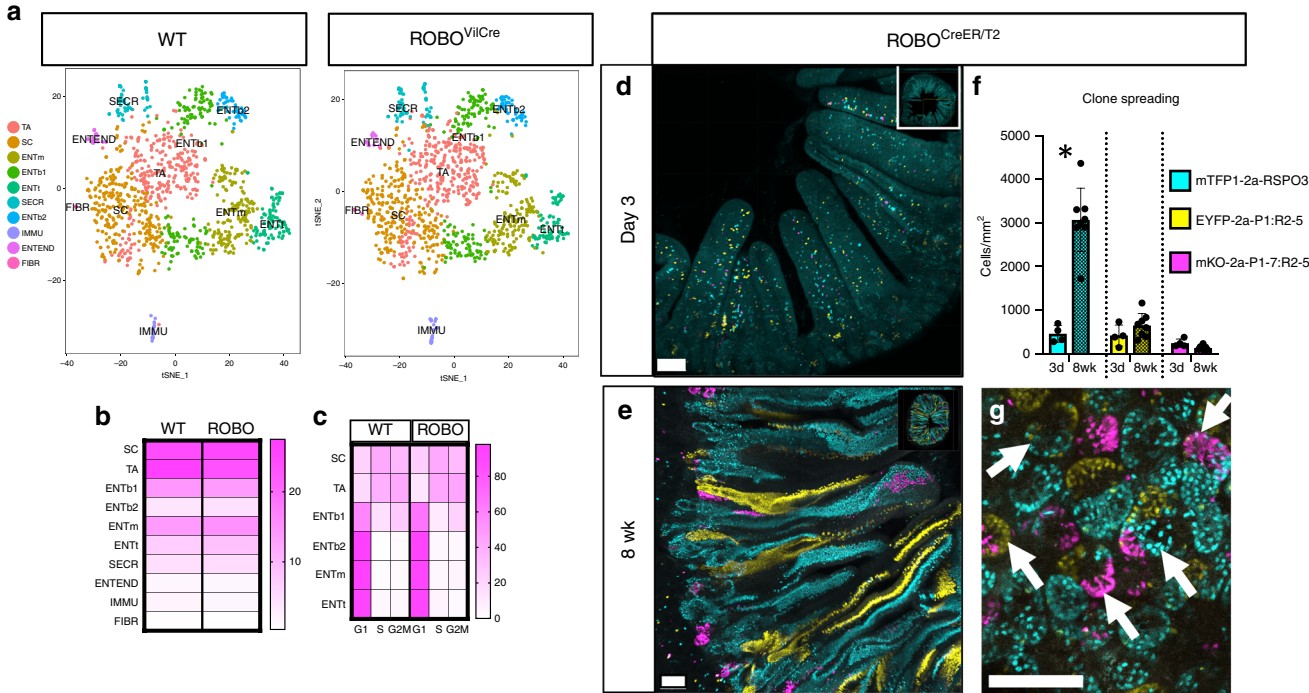

**Fig. 7** Microenvironmental oncogenesis drives heterogenous premalignant clones in adults. **a** Single cells were isolated from PND16 *WT* and *ROBO^VilCre* by fractionation to enrich for crypt epithelium. scRNAseq and t*SNE* visualization was performed for *WT* and *ROBO^VilCre* cells. (SC: stem cell, TA: transiently amplifying cell, ENT: enterocytes that included bottom (b1: S/G2M phase and b2: G1), middle (m), and top (t), and were recently described[45], SECR: secretory Paneth and Goblet cells, IMMU: immune, ENTEND: enteroendocrine, FIBR: fibroblast). For cell type markers, see also Supplementary Fig. 14. **b** Heatmap visualization of cell types present in WT and ROBO^VilCre isolates as a percentage of total cells. **c** Heat map visualization of cell cycle phase as a percentage of each cell type. **d–g** *ROBO^CreER/T2* (N = 9 mice, 6–15 weeks of age) were I.P. injected with 200 mg/kg of tamoxifen and sacrificed 3 days (N = 4) or 8 weeks later (N = 5), and the small intestine was vibratome sectioned and imaged by tiling confocal microscopy. Confocal imaged vibratome sections of *ROBO^CreER/T2* small intestines at **d** 3 days or **e** 8 weeks post tamoxifen injection. **f** Nuclei were segmented and counted and normalized to the total volume imaged. Clone spread was determined by comparing the total recombined nuclei for each fate at 8 weeks relative to day 3. The asterisk denotes statistical significance by mixed effects modelling and Sidak multicomparison correction (*RSPO3*: <0.000001, *PTPRK^e1:RSPO3^e2-5* (P1:R2-5): p = 0.31 and *PTPRK^e1-7:RSPO3^e2-5* (P1-7:R2-5): p = 0.94). **g** Evidence for attenuated crypt fixation 8 weeks post tamoxifen injection as revealed by chimeric crypts in whole-mount confocal imaging. (SEM included for each graph). Scale bars = 100 µm in **d**, **e**, **g**. Source data are provided as a "Source Data file".

independent TOP-FLASH activity. Interestingly, epithelial cadherin expression robustly inhibited the TOP-FLASH activity of ΔNßcat but not Ccat/Lef1. The in vivo subcellular localization of ΔNßcat and Ccat/Lef1 confirmed this finding by showing Ccat/Lef1 was localized to the nucleus whereas ΔNßcat was found at the lateral adherens junction. These data support the ßcatenin: Cadherin "sink" model previously proposed[37]. We conclude that the overexpression of ßcat in either MCAT or NCAT mice is unable to overcome the inhibition of *Cdh1*, further confirming the importance of *Cdh1* as a tumor suppressor[38]. Additional complexity may also be due to the compensatory and/or competitive roles of the endogenous ßcat allele in the NCAT/MCAT Crainbow models.

*MCAT* organoid experiments provided additional insight into the spread of oncogenic clones by showing that *Ccat/Lef1* expressing organoids outcompete other *ßcat* isoforms and grow in the absence of microenvironmental factors. Wnt deprivation in vivo was similarly shown in a recent study to lead to accelerated crypt fixation rates of pre-malignant ISCs[12]. Therefore, selective pressure, like injury or microenvironmental perturbations, could result in an evolutionary bottleneck, the loss of less-fit ISCs, and the dramatic expansion of a premalignant ISC population. The crypt microenvironment also maintains ISC homeostasis and is reorganized during crypt morphogenesis[41,48]. Therefore, oncogenes that restructure the crypt microenvironment can also spread rapidly in adults. *RSPO3* was an obvious

candidate due to its role as a secreted microenvironmental factor that controls crypt homeostasis and the recent discovery of its fusions with *PTPRK* in colon cancer[16,17,49]. Crainbow lineage tracing revealed that the oncogenic activation of *RSPO3* (*ROBO-Crainbow*) expands the stem cell microenvironment. This leads to rapid spread of oncogenic clones in the adult intestine—a process we term microenvironmental oncogenesis (Fig. 8b). The altered crypt microenvironment results in ISC hyperplasia, increased crypt fission, and a decreased rate of crypt fixation. The reduction in crypt fixation demonstrates that crypt fixation is not required for the optimal spread of clones. Importantly, a reduced rate of crypt fixation also resulted in heterogeneous clones that spread throughout the intestine. This phenomenon begins to explain the genetic mechanisms underlying intratumor heterogeneity[3].

Crainbow modelling was also used to compare gain-of-function activity for the *PTPRK:RSPO3* fusion isoforms found in colon cancer[17]. Endogenous *PTPRK* is expressed throughout the intestinal epithelium (Supplementary Fig. 14), whereas the expression of endogenous *RSPO3* is restricted to stromal cells residing adjacent to the crypt[50] (Fig. 5a). The *PTPRK:RPSO3* fusion results in juxtaposition of *RPSO3* into the open reading frame of *PTPRK* and should therefore drive ectopic expression of *RSPO3* and expand the microenvironment. However, each isoform may not be functionally equivalent. *WT RSPO3* and the *PTPRK^e1:RSPO3^e2-7* isoform both increased clone spreading potential, albeit at different rates. We were surprised to find an

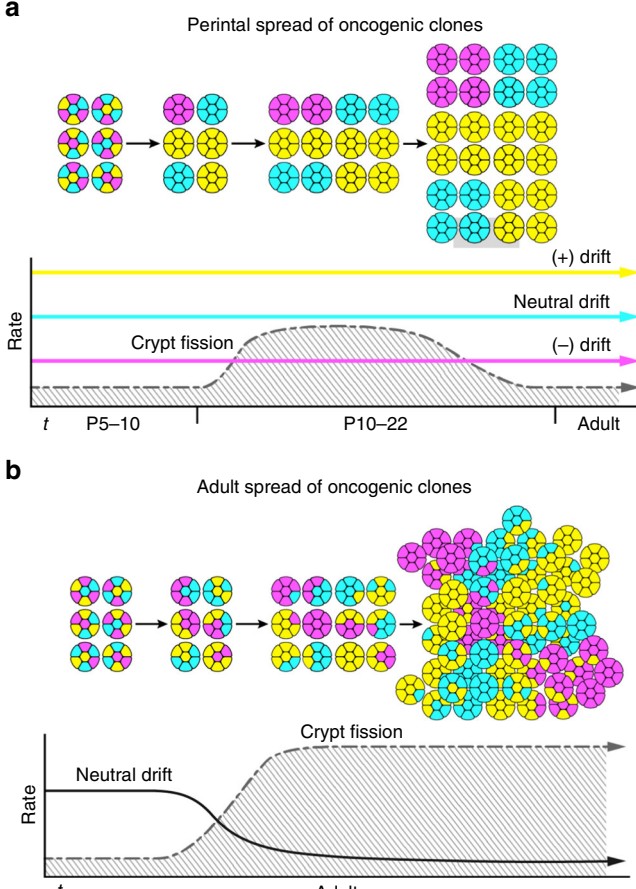

**a** Perinatal spread of oncogenic clones

(+) drift

Neutral drift

Crypt fission

(−) drift

Rate

*t* P5–10 P10–22 Adult

**b** Adult spread of oncogenic clones

Neutral drift

Crypt fission

Rate

*t* Adult

**Fig. 8** Models for the initiation and spread of oncogenic clones. **a** Early acquisition of somatic mutations during a critical period drives efficient spread of oncogenic clones before adulthood. Intestinal development selects for ISCs with increased fitness. This could bias crypt fixation (i.e., neutral drift dynamics) or crypt formation during development. Premalignant clones can then spread throughout the intestinal epithelium due to the high rate of crypt fission that is observed in mice (postnatal days 5–22, P5–22)[15] and humans (0–2 years of age)[11]. **b** Microenvironmental oncogenesis induces crypt fission and inhibits crypt fixation (i.e., neutral drift dynamics) in the adult intestine. This results in an expanded stem cell compartment and the rapid spread of heterogeneous premalignant clones in the adult intestine.

almost twofold decrease in clone spreading potential for the *PTPRK$^{e1}$:RSPO3e$^{2-7}$* isoform. *PTPRK* is a transmembrane protein tyrosine phosphatase and through homophilic interactions on adjacent cells stabilizes cell adhesion and reduces epithelial invasiveness[51]. The *PTPRK$^{e1-7}$:RSPO3$^{2-5}$* fusion retains the Ig-like domain and the fibronectin-III repeat necessary for homophilic binding. Therefore, the loss of clone spreading potential for the *PTPRK$^{e1-7}$:RSPO3$^{2-5}$* fusion could be due to potential binding with endogenous *PTPRK* and a reduction in epithelial invasiveness. RSPO3 is also a secreted protein and may have autocrine and paracrine effects on adjacent stem cells and the microenvironment.

In recent work, breast cancer evolution and functional genomics was studied using a compound cross of five mouse lines[52]. The inability to resolve multiple mutations within the same tissue and obligate laborious breeding strategies limit the utility of such an approach. The major technical benefit of our work is to directly compare oncogenic mutations in the same model system. Current models do not easily support such experiments as they

are unwieldy and not appropriately controlled. For instance, the comparison of multiple transgenic models is not recommended—transgene variegation, background strain, immunological variation, and genetic drift are just a few variables confounding the analysis. Crainbow is a transformative solution to this problem by enabling in vivo functional genomics at single cell resolution. Several imaging modalities and experimental workflows are compatible with Crainbow and together enable the direct visualization of stem cell fitness and premalignant clone competition within the intestine—together two key attributes of field cancerization. Field cancerization is a defining event of many cancers that include cancers of the breast, skin, and lung[1]. Crainbow modelling is easily adaptable and can be used to visualize the molecular and cellular attributes of field cancerization in any organ.

Several decades ago, Hubel and Wiesel demonstrated the importance of environmentally sensitive critical periods in the developing brain[53]. We broaden this paradigm to include critical periods of development sensitive to the acquisition and propagation of somatic mutations. The oncogenic activation of *RSPO3* in adult intestine reactivates the developmental critical period and facilitates the widespread expansion of premalignant clones. The accelerated spread of oncogenic clones in the adult intestine provides a cellular and molecular mechanism for driving the rapid formation of colon cancers.

## Methods

**Crainbow mouse engineering.** Infusion® (Takara, 638910) and Multisite Gateway™ Cloning (Life Technologies, 12537-100) techniques were used to engineer four customizable Gateway™ pENTR™ plasmids. All cloning steps were followed from supplier provided protocols. gBlocks® (Integrated DNA Technologies) or PCR products were used to construct positions 2–4 pENTRs with tripartite nuclear localization sequences immediately followed by a unique fluorescent protein terminated by a P2A[54] cleavage sequence and a WPRE expression enhancing element[25,55]. Position one pENTR™ was engineered using a geneblock with Mars: Sci1 FAP[56,57] followed by WPRE. Each gBlock® or PCR product was flanked by unique attB recombination sites for BP cloning to Gateway pDONR221 vectors specific to each position 1–4. The individual reactions were transformed to One Shot® Mach1 Competent cells (ThermoFisher C862003), and minipreps were Sanger sequenced to validate insert presence and sequence fidelity. Each construct after BP cloning became a pENTR™ vector with attL sites ready for LR cloning into any expression or targeting vector that contains the DEST recombination cassette. All pENTR™ vectors were whole plasmid sequenced at Massachusetts General Hospital Center for Computational & Integrative Biology DNA Core (https://dnacore.mgh.harvard.edu/new-cgi-bin/listing.action) and aligned in SnapGene (GSL Biotech LLC). Each pENTR™ was designed with a multiple cloning site containing a unique SnaBI restriction site for infusion cloning a gene of interest to pENTRs 2–4 between the P2A sequence and the WPRE element. Genes were PCR amplified or synthesized with SnaBI (ThermoFisher FD0404) compatible ends for Infusion cloning to SnaBI linearized pENTR™ vectors following supplier protocol. LR Clonase II® (Life Technologies, 11791-100) was used in an overnight reaction for Multisite Gateway™ Cloning into a ROSA26 mouse targeting vector[22,23] that we previously adapted for Gateway cloning[21]. Max Efficiency® Stbl2™ competent cells were transformed (ThermoFisher Scientific 10268-019) and grown at 30 °C to reduce spontaneous recombination. Minipreps were isolated and restriction digestion strategies were used to map potentially positive clones. Maxipreps were grown from these miniprep cultures under the same conditions, and whole plasmid sequencing was performed. ROSA-Crainbow targeting vectors were linearized by restriction digestion and then purified by Phenol:Choloroform:Isoamyl Alcohol (ThermoFisher Scientific 15593031) extraction and Ethanol precipitation. Linearized DNA was resuspended in Tris-EDTA Buffer (1×, Sigma T9285) at 1 ug/uL. Mouse embryonic stem cell gene targeting and morulae injections were performed by the Duke Transgenic Mouse facility[21].

**Mouse husbandry and genotyping.** Experiments performed in this study were conducted in accordance with an approved protocol for the ethical use of animals in research (Duke Institutional Animal Care and Use Committee protocols: A109-16-05, A079-16-04, and A034-19-02). Mice were housed in an AALAC certified specific pathogen free facility and given food and water *ad libitum*. Crainbow lines were either genotyped by PCR and by Mars1-SCi1 staining of toes (Supplementary Fig. 16). In addition to Crainbow lines, the following mouse lines were purchased from the Jackson Laboratory. ROSA FLPe—Jax-129S4/SvJaeSor-Gt(ROSA) 26Sortm1(FLP1)Dym/J (Stock No: 003946)[58], Villin Cre—Jax-B6.Cg-Tg(Vil1-cre) 997Gum/J (Stock No: 004586)[27], ROSACre$^{ER}$—JAX-B6;129-Gt(ROSA)26Sortm1

(cre/ERT)Nat/J (Stock No: 004847)[59]. Crainbow models and plasmids are available from the corresponding author upon reasonable request. Tamoxifen (Sigma, T5648) was resuspended in corn oil and delivered by intraperitoneal injection at 200 mg/kg.

**Mouse organoid culturing**. Mouse intestinal enteroids (organoids) were made according to established protocols[60–62]. Briefly, mice were sacrificed according to approved protocol and the entire intestine removed. The small intestine was cut at the intestine and cecum junction then cut in half to separate the proximal from the distal small intestine. The upper and proximal half (i.e., duodenum/jejunum) of the small intestine was flushed with 10–20 ml ice-cold PBS (Genesee 25–507) using a blunted syringe. The tissue was cut longitudinally and then cut into 2 cm pieces and washed with ice-cold PBS. Intestinal pieces were in 5 ml ice-cold 5 mM EDTA (Sigma E6758) while rotating slowly in a cold room for 30 min and then placed on ice for an additional 30 min. Tissue pieces were moved to a 15 ml tube (Supplier) containing 5 ml ice cold PBS. The tube was vigorously inverted 20–30 times, the tissue was collected and then transferred to the next 15 mL tube containing 5 ml ice-cold PBS. This process was repeated a total of five times. Ten microliters samples from each tube were assessed on a light microscope to find the fraction containing the highest density of crypts. The fraction(s) with the most crypts were filtered through a 70 µm filter (Fisher 22363548) into a BSA (10 mg/mL, Roche 10735086001)-coated conical. Isolated crypts were centrifuged at 400 g for 4 min and resuspended in ice-cold Matrigel (Corning 356237) at a concentration of 80 crypts/10 µl Matrigel. 10 µl crypt/matrigel suspension was plated per well on a 48 well cell culture plate (Corning 3548) that was prewarmed at 37 °C. The crypt/matrigel patty was polymerized by moving the plate to a 37 °C incubator for 30 min and then 200 uL base media (ENRY: EGF, Noggin, Rspondin-1, and Y-27632) was added per well. Briefly, base ENRY media was made in DMEM/F12 (Gibco 11330-032) with 1× Antibiotic/Antimycotic (Sigma A5955), 1× N-2 Supplement (Gibco 17502-048), 1× B-27 Supplement (Gibco 12587-010), 50 ng/mL EGF (Peprotech AF-100-15-100UG), 100 ng/mL Noggin (R&D Systems 1967-NG-025), 500 ng/mL Rspondin-1 (R&D Systems 4645-RS-025), and 10 uM Y-27632 (Cayman Chemical 10005583). Media was made and stored for up to one at 4 °C and changed every 2–3 days. Weekly subculture was performed by removing growth media, adding cold Gentle Dissociation Reagent (StemCell Technologies 07174), and pipetting until matrigel was dissociated from the plastic and then pipetted again until organoids were broken open to release dead cells from the organoid lumen. Dissociated organoids were centrifuged at 400 g for 4 min to pellet cells and washed once with ice cold 0.1% BSA in PBS. Organoids were pelleted by centrifugation at 400 g for 4 min and then resuspended in ice-cold matrigel. For experiments using the PORCN inhibitor Wnt-C59 (Cayman Chemical 16644), a stock solution was made at 1 mM in DMSO and then diluted to a final 10 nM in ENRY or ENY media.

**Multiprobe RNA FISH**. Advanced Cell Diagnostics RNAscope® Multiplex Fluorescent kit (323100) was used to identify mRNA targets in FFPE tissues sections from Crainbow mice. We followed the steps outlined in the user manual for FFPE sample preparation and pretreatment as well as for probe hybridization and signal development. In addition, we used dyes from the Opal® 7-color Kit (PerkinElmer, Inc NEL801001KT) for probe labelling as per company specifications. Confocal imaging was performed as described below. The following probes were used in this study: Mm-Alpi 436781, Mm-Rspo3 402011, mKO-fp 540071, EYFP-C2 312131-C2, mTFP1-C3 500271-C3, Mm-Top2a-C4 491221-C4, Mm-Lgr5-C3 312171-C3, and Mm-Reg3b-C2 478091-C2.

**Single-cell RNAseq**. We used the BioRad ddSEQ™ single-cell isolator (BioRad 12004336) for single cell capture. Briefly, intestinal crypts were isolated according to standard organoid culturing protocol. Next, isolated crypts were incubated for approximately 1 h in 0.25% Trypsin (ThermoFisher Scientific 15090046) diluted in PBS (Genesee Scientific 25–507) with 1 mg/mL Collagenase/Dispase (Sigma 10269638001), 4 uL/mL DNase I solution (ThermoFisher Scientific 90083), and 1 uM Q-VD-OPH (Cayman Chemical 15260) with pipetting every 10 min to further dissociate cell clumps. Solution was neutralized by adding three volumes of 20% FBS diluted in PBS with Ca+ and Mg+ (Sigma D8662) and passed through a 20 um mesh filter. Cells were centrifuged 400 g for 4 min and resuspended in 1 mL 0.1% BSA + 1:250 anticlumping agent (ThermoFisher Scientific 01-0057AE). Viahance™ (Biopal, Inc CP-50VQ02) reagent was used to remove dead cells and debris by adding 100 uL to cell suspension and incubating first at room temperature for 5 min then on a magnet for 10 min. The cell suspension was removed and washed in 10 ml of PBS, centrifuged, and resuspended in 0.1% BSA for counting using Trypan Blue dye and BioRad TC20™ Automated Cell Counter. Single cells were brought to a concentration of 3000/µl and libraries generated using SureCell WTA 3′ Library Prep Kit (Illumina, San Diego, CA) according to company specification. Sequencing was performed according to company specification on an Illumina NextSeq 500 in hi-output mode. Data were preprocessed in BaseSpace Sequence Hub to build gene matrix files (Illumina, San Diego CA). The gene matrix files consisting of raw counts at the gene level for each cell which was analyzed using Rstudio and Seurat(ver.2)[63]. Briefly, the gene counts for all the cells in the different conditions were combined into one matrix, normalized, and adjusted for cell cycle effects. The cell cycle phase for each cell was also determined.

Unsupervised clustering was done to separate out the cell types and markers for the cell types were identified using differential gene expression. These markers were then used for identifying the cell subpopulations such as stem cells and enterocytes.

**Luminescent signaling assays**. Experiments were performed using the HEK 293 STF stable cell line kindly provided by Dr. Jeremy Nathans and described in Xu Q, et al.[39] (PMID 15035989). STF cells were cultured in DMEM (Corning 10-013-CV) supplemented with 10% FBS (Sigma F2442) and 1% antibiotic/antimycotic (Sigma A5995) and passaged to a Poly-D-Lysine coated 96 well plate at 75,000 cells per well. We used 3 ul of Genecellin (Bulldog Bio GC1000) per 1 ug of cDNA for transfection and added the mixture during cell plating. We used 100 ng/well Renilla Luciferase (RLuc) cDNA, 10 ng/well ßcatenin Crainbow cDNAs, 90 ng/well E-cadherin (CDH1, Addgene 45769), and used empty pcDNA3.1+ to equalize total amount of cDNA across different conditions. The media was changed 24 h after transfection to MEM (ThermoFisher Scientific 51200-038) supplemented with 10 mM HEPES (ThermoFisher Scientific 15630080) and 1% GlutaMAX (Thermo-Fisher Scientific 35050061). For stimulation 10% L Cell (ATCC CRL-2648) or L Cell Wnt-3A (ATCC CRL-2647) conditioned media was added to MEM with or without R spondin-3 (R&D Systems 3500-RS) to a final concentration of 83.3 ng/ml. Plates were assayed 24 h later using Luc-Pair™ Duo-Luciferase HS Assay Kit (GeneCopeia LF005) and read on a DLR compatible luminometer.

**ßcatenin in vitro sequestration assays**. STF cells were plated on Fibronectin (Millipore FC010) coated 35 mm glass bottom dishes (MatTek P35G-0-10-C) at 1 × 10^6 cells per dish and transfected 1 µg of indicated ßcatenin Crainbow plasmid +/− E-cadherin (CDH1) and 6 µl Genecellin as above. The next day media was changed to MEM with 10% L cell conditioned media. Cells were fixed at room temperature for 15 min in 4% PFA, washed once with PBS, and permeabilized/blocked in FISHX (0.25% Triton-X diluted in 1% Fish Gelatin(Rockland MB-067-0100)) for 20 min at room temperature. Primary antibodies (Anti-HA: Abcam ab9111, Anti-V5: ThermoFisher R960-25, and Anti-Flag M2: Sigma F1804) were diluted in FISHX at 1:500 and incubated at room temperature for 1 h. Samples were washed 3 times in PBS and incubated with appropriate antibody Alexafluor™-633 (ThermoFisher Goat anti-Chicken A-21103 or anti-Mouse A-21050) diluted 1:1000 FISHX for 1 h at room temperature.

**Tissue isolation and histopathology**. For whole-mount imaging, tissues were fixed in 10% NBF overnight. A 1 cm piece of intestine was cut open lengthwise and placed villus down on a slide with Vectashield mounting medium (Vector Laboratories Cat. No: H-1000). The slide (VWR Micro Superfrost Plus, Cat. No. 48311-703) had a 22 × 22 mm coverslip (VWR micro cover glass, 22 × 22 mm, Cat. No. 48366-227) taped on both ends so a large coverslip (VWR micro cover glass, 22 × 50 mm, Cat.No. 48393-195) could span over the tissue, compressing it enough to have a flat imaging plane. NCAT whole-mount imaging was performed on unfixed and freshly isolated tissue because fixation quenched the TagBFP signal. For cross-sections (70–100 um) intestine were sectioned on a compresstome® VF-300-0Z (Precisionary). For histology, tissues were fixed in 10% NBF overnight and embedded in paraffin at the UC Davis Center for Genomic Pathology Laboratory. Tissue sections (4 µm) were stained with Mayer's hematoxylin and eosin (H&E) and whole slide imaging performed on an AT2 Scanscope (Leica Biosystems, Inc.) and stored in the Spectrum database (Leica). Images were viewed using Imagescope software (Leica).

**Tissue staining**. Antibody staining of vibratome slices was also performed for TagBFP and epitope tags as described below. For HA and V5 epitope tags, tissues were permeabilized with FISHX for 1 h, washed, and then blocked with Fc block (Innovex, NB309-5). Rb-anti-HA (Cell Signaling, Clone C29F4, Cat#3724) or Rb-anti-V5 (Cell Signaling, Clone D3H8Q, Cat#13202 s) were diluted 1:500 in FISHX and incubated overnight at 4 °C overnight, washed, and then blocked in FISHX with 5% Normal Goat Serum (NGS, Thermofisher 1610064) for 30 min. Tissues were then stained for 1.5 h with Alexa Fluor™-633 goat anti-rabbit (Thermofisher A21070) diluted 1:500 in FISHX/5%NGS. For FLAG immunostaining, Ms-IgG1-anti-FLAG (Sigma, F1804) was used at 1:250 and the protocol was similar to HA and V5 immunostaining but with the inclusion of a mouse-on-mouse blocking reagent (Jackson Immuno Research Laboratories, AffiniPure Fab Fragment Donkey Anti-Mouse IgG, 715-007-003) during tissue permeabilization and appropriate secondary antibody (Alexa Fluor™-633 goat anti-mouse IgG(g1), ThermoFisher A21126). TagBFP immunostaining was performed similar to HA and V5 staining but with a cross-reactive GuineaPig (GP) anti-TagRFP antibody (Kerafast EMU107) diluted 1:500 in FISHX and appropriate secondary antibody (Alexa Fluor™-633 goat-anti-guinea pig IgG(H + L), Thermofisher A21105). Stained tissue sections were mounted with Vectashield mounting medium (Vector Laboratories Cat. No: H-1000and imaged by confocal microscopy) on glass slides (VWR Micro Superfrost Plus, Cat. No. 48311-703) and appropriate coverslip (VWR micro cover glass, 22 × 22 mm, Cat.No.48366-227) or (VWR micro cover glass, 22 × 50 mm, Cat.No.48393-195). For in situ assessment of proliferation mice were I.P. injected with EdU (Sigma Cat. 900584, 50 mg/kg) 2 h before sacrifice. EdU staining was performed according to company protocol using a Click-iT™ Plus kit to prevent quenching of XFPs (Click-iT™ EdU Alexa Fluor™ 647 imaging kit, C10340).

**Confocal microscopy and image analysis**. Imaging was performed on a Zeiss LSM 880 confocal microscope equipped with seven lasers (405, 458, 488, 514, 561, 594, and 633 nm), 34 spectral array detectors (32 GaAsP and two PMTs). All lenses used Zeiss Plan-Apochromat, and they included 10×/0.45 NA, 20×/0.80 NA, 40×/1.20 NA Imm Corr, and 63×/1.40 NA Oil DIC. Dichroic beam splitters included: 458/514/561/633. Excitation laser and appropriate detector ranges were used for multispectral acquisition of XFPs and fluorescent stains. Images were processed and analyzed with Imaris (Bitplane), FIJI[64], and the scikit-image[65] Python package. Imaris was used to view images in 3D, adjust images for figures, and find and count spots, and in some cases count crypts. FIJI and the scikit-image package were used to build analysis pipelines for preprocessing data. Briefly, images were first pre-processed by subtracting channels where necessary to remove noise, followed by background subtraction, and gaussian or median filtering. Wholemount images were z-projected into one plane and either segmented by hand (crypt chimerism) or by thresholding and using a watershed morphological segmentation plugin (MorphoLibJ, IJBP-plugins[66]) in FIJI). Nuclei were segmented by intensity thresholding using the Spot module in IMARIS and organoids (Fig. 3c–e) were manually counted in IMARIS.

**Reporting summary**. Further information on research design is available in the Nature Research Reporting Summary linked to this article.

## Data availability

The scRNAseq datasets analyzed in the current study are available at GEO, accession GSE132402. The source data underlying Figs. 2d, g, 3d, g, 4c, f, i, j, 6f, and 7b are provided as Source Data file. The data supporting this study are available in the Article, Supplementary Information, Source Data and available from the authors upon reasonable request.

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

## Acknowledgements

The authors thank Dr. Brigid Hogan for excellent discussions, editing, and review of this paper. We also thank Dr. David Virshup for thoughtful review of this paper. The authors also thank Dr. Alan S. Waggoner for the generous gift of SCi1 fluorogen. This work was supported by NCI K12 Training grant 512-CA100639-10, NCI 1K22CA212058, Sage Biosciences IACM Supplement 3U24CA209923-01S1, Duke Surgery Start-up funds to J. C.S. and the Duke Surgery Gardner Award to J.C.S. and H.K.L. J.D.G. is supported by an NICHD 5T32HD040372. This work was also supported by NCI Grants R21CA173245 and 1R33CA191198 to L.S.B, H.K.L, and M.G.C. This work was also supported by DOD grant W81XWH-12-1-0447 to H.K.L.

## Author contributions

P.G.B. designed experiments, collected data, and analyzed data. L.K.R. designed experiments, collected data, and analyzed data. V.L. collected data and analyzed data. W.L.R. maintained the mouse colony. P.J.N. provided analysis support and instrumentation support. C.B. and M.F. conducted ES gene targeting. R.J.F. provided expertise for organoid growth. J.D.G analyzed data and interpreted data. B.R.S. designed experiments and discussed data. P.A. performed bioinformatics and scRNAseq data analysis. A.D.B. and R.D.C. performed histopathology. L.S.B., M.G.C., H.K.L., and J.C.S. conceptualized Crainbow and designed experiments. J.C.S. performed experiments, analyzed data, and wrote the paper.

## Competing interests

The authors declare no competing interests.
