## [Peer Review File · Nature Communications]

Editorial Note: Reviewer #3 was added in the third round of review as an expert in making genetic mouse models

Reviewers' comments:

Reviewer #1 (Remarks to the Author):

In the manuscript by Boone and colleagues ' A cancer rainbow mouse for visualizing functional genomics in vivo ' the authors use an innovative mouse model to study the dynamics of early tumor development in the intestine. This is achieved by parallel introducing different beta-catenin mutations in the gut coupled to specific fluorophores. It is demonstrated that although Wnt activating beta catenin mutations display enhanced expansion properties during development. However, postnatally beta catenin mutations do not promote crypt fission although in vitro an enhance clonogenicity is detected. This suggests that the compartmentalization of the adult intestine in crypts is limiting spread of premalignant clones. In the second part of the manuscript Wnt is activated more upstream using RSPO3 which in contrast resulted in marker expansion of mutant crypts presumably by modulation of the expansion rate of the gut.

The experiments are state of the art, like the high-end mouse models developed. However, at several instances the manuscript is not so straightforward to follow and not placed in the context of current literature in this area. This needs to be improved significantly.

More specific comments below:

1. The observation that there is a difference in pre- and postnatal effects of specific beta catenin mutations is an interesting observation but it is not made clear what the implications or mechanism is. It is suggested that it is related to the ability to enhance crypt fission in a permissive context (development, in vitro) but not when crypt fission is limited (adult intestine). This needs to be substantiated by specific experiments testing this hypothesis by inducing fission in adult intestine (regeneration, inflammation, Kras mutation etc)
2. Similarly, previous work indeed indicates that Wnt activation by Apc loss results in biased drift and enhanced expansion of clones within crypts in the adult intestine (e.g. Baker et al. Cell Reports 2014; Vermeulen et al Science 2013). This would be in line with the increased in vitro expansion of the Ccat/Lef1 in organoids but not in adult intestine. To place

the current work in context of this body of literature the authors should make a distinction between expansion within crypts and expansion beyond crypts (fields), and quantify both separately. This could resolve why Wnt activation (apc loss or beta cat activation) has an important impact on clonal fitness within a crypt but not beyond the crypt.

3. The nomenclature on drift and neutral drift etc. should be carefully reviewed as this is not in all instances used in line with initial key work from Benjamin Simons and coworkers. E.g. What does 'inhibition of neutral drift' (abstract) mean?

4. The putative impact of crypt fusion on the mixed crypts detected in Fig 5 should be tested (as recently described by Hugo Snippert to be much more common than previously thought).

5. Transformation in the intestine depends on a specific level of Wnt activation that is very much dependent on the position along the intestinal axis. It needs to be established what proportion of the effect between distinct mutations of beta catenin and RSPO expression / fusion on Wnt activity on specific locations in the intestine. Proximal, distal small intestine and colon.

Reviewer #2 (Remarks to the Author):

Boone et al. present a mouse model, "Crainbow", which enables the activation of 3 different mutations each of which is labelled with a fluorescent protein which can be visualized. By quantifying abundance of each fluorescent label, the effects of each of the 3 mutations on intestinal stem-cell (ISC) fitness and crypt fission can be determined in vivo. This appears to be a powerful system for quantifying how mutations impact ISC fitness and how microenvironment modulates this. The main findings of the paper seem to be (i) Ccat/Lef1 form of β catenin has increased field cancerization potential during development, but is limited in adulthood due to limited crypt fission (ii) RSPO3 expression alters the spatial arrangement of crypts and rate of crypt fission which can enable more rapid field cancerization spread.

Overall I believe this paper addresses a very interesting problem (how do pre-cancerous mutations spread in the intestine). It contains a very large amount of work the results of which will be of interest to the Nature Communications audience. However, I found the paper really tough to read: it is very densely written for the amount of work presented and seems unnecessarily short. Together with some scientific concerns, I believe this manuscript needs a revision before being considered for publication.

1. Quantification of recombination biases in the MCAT mice

The authors use total number of fluorescent cells (of a given color) as their quantitative measure for much of this work. Because the number of fluorescent cells is impacted both by recombination efficiency and by subsequent clonal expansions, the authors characterize the recombination biases in an NCAT mouse model (Figure 1). This recombination bias is then used to normalize the results in Figure 2 for the MCAT mouse (to quantify field cancerization potential of different forms of β catenin). One issue with this is that the MCAT constructs are different to the NCAT ones, so how do the authors know that the recombination biases remain the same? It is known the recombination efficiency of cre-lox depends sensitively on length, were the construct lengths designed to remain the same, to ameliorate these issues? A more convincing way to show that differences in MCAT mice are being driven by differences in field cancerization potential rather than by recombination biases would have been to sacrifice mice at earlier times and show that, at these earlier times before selection has had time to act, the biases in color usage closely reflects those observed in the NCAT mice.

2. No tracking of clones. Inability to independently quantify rates of crypt fission

A central issue in the paper is the extent to which field cancerization is driven by increased ISC fitness versus ability to increase crypt fission. For example, in Figure 2 a, b, how do we know whether the increase in Ccat/Lef1 color (yellow) is driven by increase ISC fitness or increased fission? It would have been nice to see independent estimates of these quantities. It seems standard in the field to induce fluorescence at a much lower efficiency so that when a color is present it can be confidently identified as a clone, which enables these different effects to be disentangled (since increased ISC will make many clones all of which fix in their crypt, while increased fission will create same number of clones, but clones get

larger due to expansion from fission). Did the authors try this approach? Given its ubiquity in the field, it would be nice to see a discussion of why clonal tracking was not performed.

3. Negative control for lack of field cancerization in adult MCAT mice

A key result is that field cancerization of the Ccat/Lef1 form of β catenin is limited by crypt micro-architecture: i.e. while it seems to have a fitness advantage in ISCs and fixes in crypts for often, it is then limited in its expansion due to lack of crypt fission in adulthood. The authors show this by inducing labelling in adulthood and then quantifying abundances at 3 days and 8 weeks (Figure 3). Did the authors perform the same experiment on the NCAT mice? It would have been nice to see that negative control.

4. Stagnation of field effects in adult MCAT mice

I found the claims of “stagnation” of field effects in adulthood for the Ccat/Lef1 form surprising since the plot (Figure 3b) clearly show it has an ~8-fold reduction in field cancerization potential. Why is this? It would be nice to have a more thorough discussion of this.

5. RSPO3 vs Ccat/Lef1.

If I understood things correctly, a major take home of the paper is that while some mutations (e.g. Ccat/Lef1) increase ISC fitness, their field cancerization potential are limited by low rates of crypt fission. While other mutations (e.g. RSPO3) can overcome this by increasing crypt fission rates. It would be interesting to understand better the interaction between these effects and understand in which circumstances one is more important than the other. Competing these two mutations directly in the same mice might be interesting. Did the authors do this? Also the finding that the two fusions (RSPO3-PTPRK) had less cancerization potential was not discussed. If I understood it correctly, these are known to be oncogenic. Why then do they have a lower cancerization potential relative to RSPO3?

Minor issues:

1. Referencing - seems a little odd in places e.g. ref 1 about how somatic mutations disrupt homeostasis refers to an entirely theoretical paper. In other places it seemed a bit thin on the ground.
2. ISC is not defined.
3. Outlines on figures (e.g. of the crypts in Figure 3) would have been helpful.

Response to Reviewers' comments.

Reviewer #1 (Remarks to the Author):

In the manuscript by Boone and colleagues ' A cancer rainbow mouse for visualizing functional genomics in vivo ' the authors use an innovative mouse model to study the dynamics of early tumor development in the intestine. This is achieved by parallel introducing different beta-catenin mutations in the gut coupled to specific fluorophores. It is demonstrated that although Wnt activating beta catenin mutations display enhanced expansion properties during development. However, postnatally beta catenin mutations do not promote crypt fission although in vitro an enhance clonogenicity is detected. This suggests that the compartmentalization of the adult intestine in crypts is limiting spread of premalignant clones. In the second part of the manuscript Wnt is activated more upstream using RSPO3 which in contrast resulted in marker expansion of mutant crypts presumably by modulation of the expansion rate of the gut.

The experiments are state of the art, like the high-end mouse models developed. However, at several instances the manuscript is not so straightforward to follow and not placed in the context of current literature in this area. This needs to be improved significantly.

General Response 1: Our revised manuscript is now more closely in tune with the figure and word limits for Nature Communications, and as a result is more readable. It now includes a more extensive introduction, discussion, and an expanded results section, which is accompanied by an increase of almost 20 references reflecting a more extensive critical evaluation of the current literature that more appropriately place our results in this context. Our figures have also been reformatted increasing from 5 to 8 figures in the revised manuscript. This has also enabled more clarity in the presentation of the data, the addition of new images, and allowed for more a more thoughtful description of the experimental rationale and conclusions. A point by point response to comments is included below. Where necessary we have highlighted text changes in yellow in a separate "tracked changes" document.

More specific comments below:

1.1 The observation that there is a difference in pre- and postnatal effects of specific beta catenin mutations is an interesting observation but it is not made clear what the implications or mechanism is. It is suggested that it is related to the ability to enhance crypt fission in a permissive context (development, in vitro) but not when crypt fission is limited (adult intestine). This needs to be substantiated by specific experiments testing this hypothesis by inducing fission in adult intestine (regeneration, inflammation, Kras mutation etc).

Response 1.1 Thank you for the suggestion. This question was addressed in the original manuscript in the form of the ROBO experiments that are now presented more clearly in figures 5-7. The rationale for the ROBO experiments was based upon the apparent lack of fission that occurred in MCAT mice. The lack of field cancerization in adult MCAT intestine together with the observed rapid spread in ROBO mice that is also accompanied with increased crypt fission, provides evidence for the necessity of crypt fission in the field cancerization process. Moreover, it addresses the suggested need to breed to mice with Kras mutations by offering a molecular mechanism (PTPRK/RPSO3) for driving this process at rates not yet observed in other scenarios. We have clarified the rationale for these experiments in the text that culminate with the following hypothesis as described in the subsection, **Rapid field cancerization of the adult intestine,**

“Therefore, we hypothesized that misexpression of RSPO3 and its fusion products perturb the microenvironment of the crypt and can induce field spreading by increasing crypt fission.”

1.2 Similarly, previous work indeed indicates that Wnt activation by Apc loss results in biased drift and enhanced expansion of clones within crypts in the adult intestine (e.g. Baker et al. Cell Reports 2014; Vermeulen et al Science 2013). This would be in line with the increased in vitro expansion of the Ccat/Lef1 in organoids but not in adult intestine. To place the current work in context of this body of literature the authors should make a distinction between expansion within crypts and expansion beyond crypts (fields), and quantify both separately. This could resolve why Wnt activation (apc loss or beta cat activation) has an important impact on clonal fitness within a crypt but not beyond the crypt.

Response 1.2 We have included these references in the expanded manuscript and introduce our work in this context. We now provide imaging data showing whole-mount clone fixation (i.e. within crypts) to show that expanding beyond these local crypt fields does not occur efficiently in adult MCAT mice (**Fig. 4a-c**). Our revised introduction and discussion also now include many of these important concepts

For instance, we conclude the description of the MCAT experiments as follows: “These data illustrated that the intrinsic oncogenesis of Wnt/ β cat signalling in the ISC can result in the fixation of a local field of premalignant ISCs within the intestinal crypt. This local field can spread rapidly during perinatal development but is constrained in adults by the crypt microenvironment due to the infrequent rate of crypt fission.”

1.3. The nomenclature on drift and neutral drift etc. should be carefully reviewed as this is not in all instances used in line with initial key work from Benjamin Simons and coworkers. E.g. What does ‘inhibition of neutral drift’ (abstract) mean?

Response 1.3. We have made these changes in the manuscript. For clarity we now typically use terms like “crypt fixation”. For instance:

“Intestinal crypts harbor a population of ISCs that symmetrically divide and stochastically differentiate. This results in neutral drift behavior of the ISC and crypts that are populated by a single clone of stem cells – a process also known as crypt fixation^{22,23}.”

1.4. The putative impact of crypt fusion on the mixed crypts detected in Fig 5 should be tested (as recently described by Hugo Snippert to be much more common than previously thought).

Response 1.4 This is a very inciteful suggestion. In our studies we see an increase in crypt number which would indicate that even in the presence of fusion, that crypt fission is occurring at a much higher rate. In addition, a recent report from D. Winton’s group (Cell Stem Cell, 2018) concluded that fusion had little to no effect on clone size. For our laboratory, however, the equipment and optics necessary to perform these suggested experiments are unavailable making these studies beyond our current capabilities and as a consequence beyond the scope of the current manuscript.

1.5. Transformation in the intestine depends on a specific level of Wnt activation that is very much dependent on the position along the intestinal axis. It needs to be established what proportion of the effect between distinct mutations of beta catenin and RSPO expression / fusion on Wnt activity on specific locations in the intestine. Proximal, distal small intestine and colon.

Response 1.5 We addressed this through whole mount imaging using structured illumination microscopy on a dissecting microscope. We imaged every crypt from the proximal to distal small intestine and only modest changes to crypt fitness were found across oncogenic β cat isoforms (these data are not included in this study since it is $N = 1$). Confocal analysis of $N=3$ mice demonstrated that there was a reduction in Ccat/Lef1 fitness from small intestine to colon in MCAT mice but that both the Ccat/Lef1 and $\Delta N\beta$ cat were 2x more fit than the $\Delta N\beta$ cat Δ C (**Supplementary Figure 6a-b**). This is discussed in the text of the manuscript:

“The proximal to distal axis of the intestine differs in Wnt/ β catenin signaling potential^{32,33}. We found that $\Delta N\beta$ cat Δ C was also disadvantaged in the colon but that the effects of Ccat/Lef1 were mitigated and was now more similar to $\Delta N\beta$ cat (**Supplementary Figure 6a-b**).”

We did not perform fitness calculations for ROBO as instead we focused on crypt fixation. In the revision we have included wholemount images of ROBO colon to illustrate the similar inhibition of crypt fixation that was observed in small intestine (**Fig. 6f-g**).

Reviewer #2 (Remarks to the Author):

Boone et al. present a mouse model, “Crainbow”, which enables the activation of 3 different mutations each of which is labelled with a fluorescent protein which can be visualized. By quantifying abundance of each fluorescent label, the effects of each of the 3 mutations on intestinal stem-cell (ISC) fitness and crypt fission can be determined in vivo. This appears to be a powerful system for quantifying how mutations impact ISC fitness and how microenvironment modulates this. The main findings of the paper seem to be (i) Ccat/Lef1 form of β catenin has increased field cancerization potential during development, but is limited in adulthood due to limited crypt fission (ii) RSPO3 expression alters the spatial arrangement of crypts and rate of crypt fission which can enable more rapid field cancerization spread.

Overall I believe this paper addresses a very interesting problem (how do pre-cancerous mutations spread in the intestine). It contains a very large amount of work the results of which will be of interest to the Nature Communications audience. However, I found the paper really tough to read: it is very densely written for the amount of work presented and seems unnecessarily short. Together with some scientific concerns, I believe this manuscript needs a revision before being considered for publication.

General Response 2: See **General Response 1** regarding a similar readability concern raised by reviewer 1.

1. Quantification of recombination biases in the MCAT mice

The authors use total number of fluorescent cells (of a given color) as their quantitative measure for much of this work. Because the number of fluorescent cells is impacted both by recombination efficiency and by subsequent clonal expansions, the authors characterize the recombination biases in an NCAT mouse model (Figure 1). This recombination bias is then used to normalize the results in Figure 2 for the MCAT mouse (to quantify field cancerization potential of different forms of β catenin). One issue with this is that the MCAT constructs are different to the NCAT ones, so how do the authors know that the recombination biases remain the same? It is known the recombination efficiency of cre-lox depends sensitively on length, were the construct lengths designed to remain the same, to ameliorate these issues? A more

convincing way to show that differences in MCAT mice are being driven by differences in field cancerization potential rather than by recombination biases would have been to sacrifice mice at earlier times and show that, at these earlier times before selection has had time to act, the biases in color usage closely reflects those observed in the NCAT mice.

Response 2.1 Position 1 of NCAT and MCAT are the same isoform of β cat ($\Delta N\beta$ cat) and same size. Therefore, we do not expect to see significant changes to bias in NCAT vs MCAT mice. In addition, we have clarified MCAT^{CreER/T2} experiments (**Fig.4**) in which we utilize a tamoxifen approach to induce Crainbow activation and then image at 3 days and 8 weeks later. In so doing and as suggested, **Figure 4a/c** shows that at day 3 we can observe a similar positive bias for position 1 (about 2-fold) and negative bias for positions 2 and 3. At this early timepoint, any differences in cell number are due to recombination bias since the ISC has not had time to clonally expand and has likely only proliferated once. This bias is effectively what we had observed in our control NCAT lines and suggests that while construct length can play a role in recombination efficiency the construct length is similar enough in MCAT/NCAT mice to enable predictable bias normalization.

We have included the following statement in the text to describe the normalization rationale: “Since the same $\Delta N\beta$ cat allele was used in Position 1 of NCAT and MCAT mice, we normalized to the positional recombination bias calculated in NCAT mice to obtain values of relative field cancerization potential in MCAT mice. ”

2. No tracking of clones. Inability to independently quantify rates of crypt fission

A central issue in the paper is the extent to which field cancerization is driven by increased ISC fitness versus ability to increase crypt fission. For example, in Figure 2 a, b, how do we know whether the increase in Ccat/Lef1 color (yellow) is driven by increase ISC fitness or increased fission? It would have been nice to see independent estimates of these quantities. It seems standard in the field to induce fluorescence at a much lower efficiency so that when a color is present it can be confidently identified as a clone, which enables these different effects to be disentangled (since increased ISC will make many clones all of which fix in their crypt, while increased fission will create same number of clones, but clones get larger due to expansion from fission). Did the authors try this approach? Given its ubiquity in the field, it would be nice to see a discussion of why clonal tracking was not performed.

Response 2.2 These experiments were present in the original submission but were obscured by the readability concerns. We have resolved this by providing clonal tracking data in the expanded Figure 4 to illustrate that polyclonal crypts become “fixed” in the crypt as single colors/lineages over time. In other words, as was suggested, many clones are formed that fix in the crypt and we do not see evidence for the same number of clones getting larger due to crypt fission/expansion. Qualitative evidence also shows that we do not see large patches of single-color fixed crypts, which also suggests that significant increases in crypt fission and therefore clone/patch size are not observed (Fig. 4a,b). The lack of crypt fission in adult intestine constrained the spread of fields. This hypothesis was formally tested in ROBO Crainbow experiments by inducing crypt fission and driving rapid field cancerization.

3. Negative control for lack of field cancerization in adult MCAT mice

A key result is that field cancerization of the Ccat/Lef1 form of β catenin is limited by crypt micro-architecture: i.e. while it seems to have a fitness advantage in ISCs and fixes in crypts for often, it is then limited in its expansion due to lack of crypt fission in adulthood. The authors show this

by inducing labelling in adulthood and then quantifying abundances at 3 days and 8 weeks (Figure 3). Did the authors perform the same experiment on the NCAT mice? It would have been nice to see that negative control.

Response 2.3 We did not perform this experiment since the MCAT mice express the control $\Delta N\beta\text{cat}$ isoform in addition to the two experimental isoforms (Ccat/Lef1 and $\Delta N\beta\text{cat}\Delta C$). In the MCAT, studies $\Delta N\beta\text{cat}$ does not cause an increase in relative field spreading in this experiment (now **figure 4**). This points to the utility of Crainbow since each line can be used as its own control.

4. Stagnation of field effects in adult MCAT mice

I found the claims of “stagnation” of field effects in adulthood for the Ccat/Lef1 form surprising since the plot (Figure 3b) clearly show it has an ~8-fold reduction in field cancerization potential. Why is this? It would be nice to have a more thorough discussion of this.

Response 2.4 The goal of this experiment was to determine if any of the MCAT oncogenes could increase field spread. Clearly none of these mutations were able to do so, which is why we used the word “stagnation”. We have tempered the conclusion in the results to now read:

“This analysis revealed that recombined cells did not increase from day three to eight weeks for any of the oncogenic βcat isoforms and suggesting again that somatic field spreading was constrained to local fields within a crypt (**Fig. 4d**).”

We also agree that the observed decrease in Ccat/Lef1 fields is surprising. This an interesting finding but it should be noted that this is a trend toward a reduction and is not statistically significant due to the variability at day 3. However, we agree that this must be addressed and we have included the following interpretation:

“Surprisingly there was an apparent but statistically insignificant retraction of Ccat/Lef1 fields confirming previous reports that even ISCs with potent somatic mutations are still subject to stochastic loss and replacement by neighboring wild-type stem cells¹¹”.

5. RSPO3 vs Ccat/Lef1.

If I understood things correctly, a major take home of the paper is that while some mutations (e.g. Ccat/Lef1) increase ISC fitness, their field cancerization potential are limited by low rates of crypt fission. While other mutations (e.g. RSPO3) can overcome this by increasing crypt fission rates. It would be interesting to understand better the interaction between these effects and understand in which circumstances one is more important than the other. Competing these two mutations directly in the same mice might be interesting. Did the authors do this? Also the finding that the two fusions (RSPO3-PTPRK) had less cancerization potential was not discussed. If I understood it correctly, these are known not to be oncogenic. Why then do they have a lower cancerization potential relative to RSPO3?

Response 2.5

Question 1: The suggestion to cross Crainbow mice is an excellent one. In fact, we have Crainbow mice that target the Rspo3 receptor (Lgr5) that are not yet fully described and we are currently crossing these to ROBO mice to test for ligand:receptor interactions. We have not attempted this for MCAT:ROBO. In both cases, these experiments are not very straightforward due to the use of the same color palette that is used in each Crainbow mouse line. We are

currently developing RNA FISH protocols to discriminate each Crainbow line but it is beyond the scope of the current manuscript.

Question 2: In our expanded manuscript we devote a paragraph in the discussion for describing how RSPO3 and its fusions could have discrete signaling modalities within the intestinal stem cell compartment. The paragraph is included below:

“Crainbow modelling was also used to compare gain-of-function activity for the *PTPRK:RSPO3* fusion isoforms found in colon cancer⁴¹. Endogenous *PTPRK* is expressed throughout the intestinal epithelium (**Supplementary Figure 14**) whereas the expression of endogenous *RSPO3* is restricted to stromal cells residing adjacent to the crypt⁵¹ (**Fig. 5A**). The *PTPRK:RSPO3* fusion results in juxtaposition of *RSPO3* into the open reading frame of *PTPRK* and should therefore drive ectopic expression of *RSPO3* and expand the microenvironment. However, each isoform may not be functionally equivalent. The *WT RSPO3* and the *PTPRK^{e1}:RSPO3^{e2-7}* both increased field cancerization, albeit at different rates. We were surprised to find an almost two-fold decrease in field cancerization potential for the *PTPRK^{e1}:RSPO3^{e2-7}* isoform. *PTPRK* is a transmembrane protein tyrosine phosphatase and through homophilic interactions on adjacent cells stabilizes cell adhesion and reduces epithelial invasiveness⁵². The *PTPRK^{e1-7}:RSPO3²⁻⁵* fusion retains the Ig-like domain and the fibronectin-III repeat necessary for homophilic binding. Therefore, the loss of field cancerization potential for the *PTPRK^{e1-7}:RSPO3²⁻⁵* fusion could be due to its ability to potentially bind to endogenous *PTPRK* and reduce epithelial invasiveness.”

Minor issues:

1. Referencing - seems a little odd in places e.g. ref 1 about how somatic mutations disrupt homeostasis refers to an entirely theoretical paper. In other places it seemed a bit thin on the ground.

We have included almost 20 more references to more thoroughly and thoughtfully review the literature in the context of our data.

2. ISC is not defined.

Apologies for this mistake. This has been fixed.

3. Outlines on figures (e.g. of the crypts in Figure 3) would have been helpful.

We have included a few representative outlines in Figure 4 (Previously Figure 3).

Reviewers' comments:

Reviewer #1 (Remarks to the Author):

The revised version of the manuscript by Boone and colleagues has been improved at several instances but yet in other areas is not sufficiently clear to warrant publication.

Like the initial version the manuscript is still very difficult to follow given the multitude of different models and experimental setups. In addition the relation to previous literature in this area has been satisfyingly addressed.

For example, I requested an analysis that disentangles the clonal expansion effect of the various B-cat mutations within the crypts and beyond the crypts but I fail to see the answer in the current manuscript. As previously it was demonstrated that oncogenic activation of the Wnt pathway results in more rapid fixation of clones 'within' the crypt. It is important to demonstrate that this is captured in the current model at least for that beta catenin mutations that activates Wnt. This could be done by quantifying the clone sizes of the various mutations and controls, preferably at different time points following or at least at one representative time point. Also see my initial comments.

Furthermore, it is suggested that Wnt inhibition enhances the efficacy of oncogenic spread in vivo. Also here it is unclear what part of this effect is because of accelerated intracryptal expansion leading to fixation and what part is due to enhanced fission leading to field cancerization.

I strongly recommend these issues to be addressed as currently it is unclear how the presented observations fit into the existing literature. In addition the authors might consider simplifying their nomenclature as the current set of model and mutation names is far from intuitive and difficult to understand even for somebody that is very much involved in this field.

Reviewer #2 (Remarks to the Author):

The authors have addressed my concerns and I find the manuscript much improved. I recommend publication.

Response to Reviewers' comments.

Reviewer #1 (Remarks to the Author):

Comment1: The revised version of the manuscript by Boone and colleagues has been improved at several instances but yet in other areas is not sufficiently clear to warrant publication.

Like the initial version the manuscript is still very difficult to follow given the multitude of different models and experimental setups. In addition, the relation to previous literature in this area has been satisfyingly addressed.

Response1: Thank you for your helpful comments. Communicating the complexity of our model system has been a challenge. As per your suggestions, we have added significant detail to the introduction to better place our findings in context with previous literature. This more effectively communicates the significance of our finding that *RSPO3* and its isoforms can induce field cancerization in a few weeks. The improvements to the manuscript also include a table succinctly describing the genetics of Crainbow lines and key observations from each study (Table1). The table is included in the text and below. We hope this helps to simplify the presentation of the mouse models and their usage. We will ask that the “Map” have a hyperlink to the supplementary figure so that the reader can seamlessly access the detailed view of each transgenic mouse line.

Table 1 Overview of cancer rainbow (Crainbow) mouse lines

Symbol	Name	Oncogene and Fluorescent Barcode*	Key Observations	Map
NCAT	N-terminal truncated β catenin	Cre off \rightarrow Mars1-FAP Cre on \rightarrow [ 1. $\Delta N\beta$Cat 2. $\Delta N\beta$cat 3. $\Delta N\beta$cat 	The positional recombination efficiency of Crainbow is biased 2.3-fold at position 1, 0.36-fold at position 2, and 0.32-fold at position 3.	S4
MCAT	Multiple isoforms of β catenin	Cre off \rightarrow Mars1-FAP Cre on \rightarrow [ 1. $\Delta N\beta$Cat 2. Ccat/Lef1 3. $\Delta N\beta$catΔC 	Field cancerization occurs rapidly during perinatal development but is constrained by intestinal epithelial homeostasis in adult.	S5
ROBO	RSPO3 Crainbow	Cre off \rightarrow Mars1-FAP Cre on \rightarrow [ 1. RSPO3 2. PTPRK^{e1}:RSPO3^{e2-5} 3. PTPRK^{e1-7}:RSPO3^{e2-5} 	RSPO3 oncogenes expand the crypt microenvironment to induce crypt fission and promote rapid field cancerization of the adult epithelium.	S7

*Brief description of oncogenes and Crainbow lines. (NCAT) $\Delta N\beta$ cat is based upon the previously described N-terminal truncation mutant and escapes degradation to increase Wnt-signaling²⁶. NCAT mice express $\Delta N\beta$ cat in positions 1-3 and are used to calculate positional bias in Crainbow transgenes. (MCAT) MCAT mice express three isoforms of β cat that act as oncogene prototypes. $\Delta N\beta$ cat was directly compared to β cat isoforms that possess an increased Wnt-signaling potential (*Ccat/Lef1*)^{34,35} or a decreased Wnt-signaling potential ($\Delta N\beta$ cat Δ C)³⁴⁻³⁶. MCAT mice are used to test the comparative ability of Crainbow modelling and measure the field cancerization potential of three oncogenes. (ROBO) *RSPO3* is expressed by the crypt microenvironment and controls stem cell homeostasis⁵¹. The recently described oncogenic fusions of *PTPRK^{e1}:RSPO3^{e2-5}* and *PTPRK^{e1-7}:RSPO3^{e2-5}* are known drivers of colorectal cancer¹⁶⁻¹⁸. ROBO mice are used to assess how oncogenic activation of microenvironmental cues could also be potent drivers of rapid field cancerization in the adult intestinal epithelium. Oncogenes are pseudocolored to match the fluorescent protein barcode.

Comment 2: For example, I requested an analysis that disentangles the clonal expansion effect of the various B-cat mutations within the crypts and beyond the crypts but I fail to see the answer in the current manuscript.

As previously it was demonstrated that oncogenic activation of the Wnt pathway results in more rapid fixation of clones 'within' the crypt. It is important to demonstrate that this is captured in the current model at least for that beta catenin mutations that activates Wnt. This could be done by quantifying the clone sizes of the various mutations and controls, preferably at different time points following or at least at one representative time point. Also see my initial comments.

Response 2: Thank you for your insight. We apologize for the lack of clarity and for not including the requested experiment. We now provide two separate adult experiments in MCAT mice to answer this important question. **Fig. 4a-c** provides data from a tamoxifen experiment in which we assess "beyond crypt" field cancerization. Our data illustrate that significant beyond crypt field spreading does not occur, especially in comparison to ROBO mice (**Fig. 7d-g**). As per your advice, **Fig. 4d-f** includes the results from a new experiment to investigate "within the crypt" fixation. Our findings show that $\Delta N\beta\text{cat}$ modestly increases fixation while $Ccat/Lef1$ and $\Delta N\beta\text{cat}\Delta C$ is modestly reduced. Vermeulen et.al (Science, 2013) have previously demonstrated intestinal homeostasis suppresses the fixation of somatic mutations (including APC-loss). Therefore, our data largely confirm the work of Vermeulen et al.

Comment 3: Furthermore, it is suggested that Wnt inhibition enhances the efficacy of oncogenic spread in vivo. Also here it is unclear what part of this effect is because of accelerated intracryptal expansion leading to fixation and what part is due to enhanced fission leading to field cancerization.

Response 3: We apologize if this was a source of confusion. The finding that WNT inhibition enhances the efficacy of oncogenic spread was previously demonstrated by Huels et. al. (Nature Communications 2018) and is cited accordingly in the discussion and now also in the introduction. MCAT organoid studies provide a reasonable explanation for the finding of Huels, et.al. by showing that $Ccat/Lef1$ ISCs can survive the loss of Wnt/RSPO factors whereas $\Delta N\beta\text{cat}$ and $\Delta N\beta\text{cat}\Delta C$ do not. Therefore, selection of the fittest ISC can occur within a few days resulting in $Ccat/Lef1$ positive epithelium. These data illustrate how selective pressure (i.e. injury or drug therapy) could result in the rapid propagation of somatic mutations. Future studies include breeding our mice to βcat KO mice and studying intestinal development to determine how these fields spread during development (i.e. crypt fission, crypt fixation, or crypt morphogenesis).

Comment 4: I strongly recommend these issues to be addressed as currently it is unclear how the presented observations fit into the existing literature. In addition, the authors might consider simplifying their nomenclature as the current set of model and mutation names is far from intuitive and difficult to understand even for somebody that is very much involved in this field.

Response 4: Thank you again for your helpful critiques and thoughtful advice. We have made these corrections and included the requested experiment as requested. The nomenclature remains the same because we do think it is the best way to describe our novel mouse models. As we indicated above, we have succinctly described each model system in table form and hope that this helps increase the clarity of our manuscript.

Reviewer #2 (Remarks to the Author):

Comment 1: The authors have addressed my concerns and I find the manuscript much improved. I recommend publication.

Response 1: Thank you.

Reviewers' comments:

Reviewer #1 (Remarks to the Author):

My comments have been addressed.

Reviewer #3 (Remarks to the Author):

Snyder and colleagues present elegant mouse genetics and stunning microscopy of the mouse small intestine in their study on the impact of mutations on clonal fitness. However, the absolute necessity of rainbow for studying clonal competition remains questionable, as individual marked mutations would be similarly effective. Although it would require 3 mouse models rather than 1, the authors fail to convince why the study on 3 oncogenes in 1 intestine is of utmost importance (apart perhaps from studying competition between mutants, which is clinically not relevant). In general there are serious concerns regarding the technical aspects of the study and interpretation of the data.

First, field cancerization describes the clonal expansion of a population of cells that is genetically altered and as a result predisposes for cancer development, yet is morphologically indistinguishable from its normal counterparts. Frequently it concerns reduced expression (epigenetic) or mutated versions of DNA repair enzymes. Studying field cancer clones that show no morphological change, argues for lineage tracing techniques that can co-express (fluorescent) markers with any type of mutant protein that is potentially involved in driving accelerated clonal competition. In this manuscript, I am a) not convinced whether the beta-catenin mutants are oncogenic at all, while b) the R-spondin fusions drive adenoma development by themselves rather than pre-dispose to cancer development that requires a second hit. As such, both mouse models do not involve field cancerization studies as implicated throughout the text, but rather study adenoma development.

Major comments:

1) The authors study three different versions of Beta-catenin. First, the rationale for choosing these three is not adequately explained. There is no clinical relevance in studying these mutants. Second, I am questioning whether the used deltaN mutant versions of Beta-catenin are truly oncogenic. There is no information in the manuscript how the deltaN-mutant is constructed (which amino acids are truly missing?). The only reference in the manuscript points to the delta exon 3 mutant made by Mark Taketo, but "delta exon 3" is not similar as a deltaN truncated beta-catenin. In particular, the delta exon 3 Beta-catenin is highly oncogenic with ~3000 intestinal tumors visible by week 3 after birth (moreover, this mouse model is widely used and phenotypically confirmed by many labs). Here however, there seems to be no signs of intestinal tumor development for the NCAT mouse, questioning the oncogenic potential of the delta N mutant Beta-catenin and relevance of studying this mutant. Same holds true for the Lef1 fusion, as this is an artificial fusion mutant made by Grosschedl and colleagues to confirm that transcriptional activity of Lef1 is mediated by its binding partner Beta-catenin. The deltaN/ deltaC mutant lacks all transactivation domains and is most likely not capable of driving Wnt/beta-catenin transcriptional activity.

Ultimately, it will be essential that the authors show signs of adenoma development given enough time to develop. In such a scenario, is the adenoma development initiated by a 'second hit' within a larger 'field' of mutant beta-catenin underscoring the pre-disposition towards cancer development?

2) Field cancerization involves somatic mutations that impact the competitive fitness of the cell involved. As these are rare events, the fitness bias that is of importance lies between the mutant cell and its normal neighbours. For the intestinal epithelium that is characterized by high-turnover,

the central players are the stem cells that reside at the bottom of crypts. Therefore, mapping clonal competition involves two stages. First, the competitive bias of the mutant stem cell towards neighboring stem cells (scoring should be the fraction of the stem cell compartment within a crypt that has been taken over by the mutant, see many papers by the Winton lab). Second, upon crypt fixation, the pace of crypt fission (accelerated, neutral, etc) needs to be characterized as this underlies the mechanism by which mutant tissue can spread (concerns the scoring of the number of fully fixed crypts that encompass 1 clone).

Counting the number of cells within a clone is not informative, as most cells within a clone are rapidly turned-over in the intestine. The focus must be on the number of stem cells (fraction of partial/full crypts) that fuel a clone.

3) Studying field cancerization using crainbow requires sporadic activation in adult epithelium. All experiments using constitutive Cre activities with onsets during embryogenesis (VillinCre) are not suitable for interpretation. First, (see point 1), I am questioning the oncogenic potential of the deltaN-beta catenin mutants. Second, there is no competition between mutant cells and normal neighbours. As all cells contain recombined crainbow alleles, there is only competition created between mutant clones. This has no relevance towards patients, as field cancerization are sporadic events. The only competitive bias that matters is between a mutant clone and normal neighbouring cells.

4) Rosa-CreERT2 recombines the crainbow alleles in all cell types. Although most mutant cells are lost due to self-renewal of the epithelium, two types of clones resist. First, stem cell driven clones that are of interest to the study. The second however, are relatively long-lived (~up to 8wks) Paneth cells at crypt bottoms. These cells complicate analysis. As Paneth cells are post-mitotic, EdU pulses might be used to discriminate them from cycling stem cells. Alternatively (best option) is to use Lgr5-EGFP-Ires-CreERT2 knock-in mice to initiate sporadic clonal competition (crainbow activation) in stem cells.

5) The work concerning R-spondin fusions is more interesting as it involves patient mutations. However, since these mutant cells establish neoplasias by themselves rather than pre-disposition, it disqualifies itself from a field cancerization study. Nevertheless, studying adenoma development driven by R-spondin fusions is still interesting by itself.

Again, expression of R-spondin fusions from embryonic stages onwards is not interesting when it involves near-complete recombination of all cells in the epithelium.

Adenoma development can be studied upon sporadic activation in adult epithelia as shown in the manuscript. However, clonal competition between different mutant clones is complicated in the case of R-spondin. Foremost, just like wild-type R-spondin, the described R-spondin fusions have been reported to be secreted. Therefore, the mutant phenotype is not intrinsic to the clone itself, but will also influence neighboring cells (potentially neighboring clones), severely complicating its analysis.

Minor comments:

The word-art is very annoying. Please remove black outlines that surround coloured font.

Reviewer #3 (Remarks to the Author):

We appreciate your thoughtful suggestions and insight. The revision has given us the opportunity to sharpen our manuscript. These were indeed important issues to clarify and we are thankful for your excellent advice. We have included a point-by-point response to each comment below.

General Comment 1. Snyder and colleagues present elegant mouse genetics and stunning microscopy of the mouse small intestine in their study on the impact of mutations on clonal fitness. However, the absolute necessity of Crainbow for studying clonal competition remains questionable, as individual marked mutations would be similarly effective. Although it would require 3 mouse models rather than 1, the authors fail to convince why the study on 3 oncogenes in 1 intestine is of utmost importance (apart perhaps from studying competition between mutants, which is clinically not relevant).

General Response 1. *Crainbow fulfills an important need facing the scientific community, namely how do we empirically compare the activity of oncogenes in vivo with reliability and robustness. We have included the following statement in the discussion, “The major benefit of our work is to directly compare oncogenic mutations in the same model system. Current models do not easily support such experiments as they are unwieldy and not appropriately controlled. For instance, the comparison of multiple transgenic models is not recommended – transgene variegation, background strain, immunological variation, and genetic drift are just a few variables confounding the analysis.”*

This is especially important for comparing the potential of oncogenic variants (i.e. R-spondin fusions etc.). Most models also lack fluorescent lineage tags or use tags which are not easily imaged. Breeding three lines together and a Cre-recombinase also results in extensive compound crossing. Absent the utilization of orthogonal site-specific recombinases each oncogene would also be active in the same cell. Crainbow provides a solution for each of these problems in a single genetically tractable system. Future studies will use Crainbow models to test preclinical therapy/immunotherapy on oncogenic variants (R-spondin Crainbow mice and other models yet to be described). In other cases, we envision Crainbow will be used to directly compare tumorigenic potential and tumor evolution.

General Comment 2. In general, there are serious concerns regarding the technical aspects of the study and interpretation of the data. First, field cancerization describes the clonal expansion of a population of cells that is genetically altered and as a result predisposes for cancer development, yet is morphologically indistinguishable from its normal counterparts. Frequently it concerns reduced expression (epigenetic) or mutated versions of DNA repair enzymes. Studying field cancer clones that show no morphological change, argues for lineage tracing techniques that can co-express (fluorescent) markers with any type of mutant protein that is potentially involved in driving accelerated clonal competition. In this manuscript, I am a) not convinced whether the beta-catenin mutants are oncogenic at all, while b) the R-spondin fusions drive adenoma development by themselves rather than pre-dispose to cancer development that requires a second hit. As such, both mouse models do not involve field cancerization studies as implicated throughout the text, but rather study adenoma development.

Response 2a: *Please see response in Major response 1a below. In short, oncogenic signaling potential has been confirmed in TOPFLASH assays and in organoid culture (Figure 3/4).*

Response 2b: *Graham and colleagues recently and extensively reviewed the field cancerization literature in Nature Reviews Cancer, 2018 (PMID 29217838). The empirical observation is that field cancerization occurs with or without noticeable morphological change. “The field may or may not exhibit morphological change (for example, cancerized cells may look normal or could exhibit dysplasia)” and again, “The size of the cancerized field varies substantially, and a cancerized area of tissue will have a microscopic morphology classified as noncancerous, that is, normal, hyperplasia, metaplasia or dysplasia.” β catenin Crainbow mice (MCAT/NCAT) model field cancers without morphological change. In contrast, Rspodin Crainbow mice model field cancers with a range of phenotypes –normal and progressing to hyperplasia and eventual adenoma-like phenotypes occurring much later. From this perspective, the Crainbow mice described in our manuscript are excellent tools for studying the initiation of field cancers and their progression. We include two additional sentences in the introduction that summarizing an operational definition of field cancers “Field cancers are operationally defined by Graham and colleagues as “a group of cells that are considered to be further along an evolutionary path towards cancer.”¹ Fields can present with or without morphologic pathology.”*

Major Bulleted comments:

1a) The authors study three different versions of Beta-catenin. First, the rationale for choosing these three is not adequately explained. There is no clinical relevance in studying these mutants. Second, I am questioning whether the used deltaN mutant versions of Beta-catenin are truly oncogenic. There is no information in the manuscript how the deltaN-mutant is constructed (which amino acids are truly missing?). The only reference in the manuscript points to the delta exon 3 mutant made by Mark Taketo, but "delta exon 3" is not similar as a deltaN truncated beta-catenin. In particular, the delta exon 3 Beta-catenin is highly oncogenic with ~3000 intestinal tumors visible by week 3 after birth (moreover, this mouse model is widely used and phenotypically confirmed by many labs). Here however, there seems to be no signs of intestinal tumor development for the NCAT mouse, questioning the oncogenic potential of the delta N mutant Beta-catenin and relevance of studying this mutant. Same holds true for the Lef1 fusion, as this is an artificial fusion mutant made by Grosschedl and colleagues to confirm that transcriptional activity of Lef1 is mediated by its binding partner Beta-catenin. The deltaN/deltaC mutant lacks all transactivation domains and is most likely not capable of driving Wnt/beta-catenin transcriptional activity.

Response 1a. *Thank you for the excellent suggestions. First, we are more transparent in the use of the β cat isoforms as prototypes to validate Crainbow studies and have updated the text accordingly: “Genetic inactivation of adenomatous polyposis coli (APC) potentiates Wnt signalling and is a known driver of colon cancer³⁰⁻³². β catenin is downstream signaling effector and a prime candidate for increasing stem cell fitness and initiating field cancers. Several isoforms of β catenin have been used previously to study Wnt-signaling.... These somatic mutations vary in their Wnt-signaling aptitude and are excellent prototypes for validating Crainbow as a robust system that quantifies field cancerization potential in vivo.”*

We apologize for neglecting to include the amino acid positions of truncated β catenin mutations in NCAT and MCAT mice. Specifically, $\Delta N\beta$ cat encodes aa. 80-781 thereby modeling the exon 3 deletion made by Mark Taketo’s lab (exon 3 encodes aa. 5-79). Ccat/Lef1 encodes the transactivation domain of β cat (aa.693-781) fused in-frame with N-terminally truncated Lef1 (aa.57-398). Lastly, $\Delta N\beta$ cat Δ C encodes aa.80-693. Table 1 has been updated accordingly to include these annotations. In addition, we have also included Supplementary Files 2-4 (genbank format) of annotated sequence files for NCAT, MCAT, and ROBO transgenes.

Lastly, the lack of adenoma formation in the NCAT and MCAT mouse was also surprising to us. We have updated the discussion appropriately. “The lack of adenomas in β cat Crainbow mice (NCAT or MCAT) was surprising – especially since Harada and colleagues have shown previously that extensive intestinal polyps form when the N-terminal domain of endogenous β catenin is deleted. Our in vitro signaling assays confirmed $\Delta N\beta$ cat and Ccat/Lef1 isoforms were oncogenic as measured by Wnt/Rspondin independent TOP-FLASH activity. Interestingly, epithelial cadherin expression robustly inhibited the TOP-FLASH activity of $\Delta N\beta$ cat but not Ccat/Lef1. The in vivo subcellular localization of $\Delta N\beta$ cat and Ccat/Lef1 confirmed this finding by showing Ccat/Lef1 was localized to the nucleus whereas $\Delta N\beta$ cat was found at the lateral adherens junction. These data support the β catenin:Cadherin “sink” model previously proposed³⁷. We conclude that the overexpression of β cat in either MCAT or NCAT mice is unable to overcome the inhibition of Cdh1, further confirming the importance of Cdh1 as a tumor suppressor³⁸.”

1b) Ultimately, it will be essential that the authors show signs of adenoma development given enough time to develop. In such a scenario, is the adenoma development initiated by a 'second hit' within a larger 'field' of mutant beta-catenin underscoring the pre-disposition towards cancer development?

Response 1b. *Thank you for the excellent suggestions. We have aged NCAT and MCAT mice in the colony and to date have not found evidence for the development of adenomas. We are currently pursuing studies using chemical inducers and inflammatory stimuli to assess the second hit potential within β cat induced fields. A caveat of the MCAT/NCAT studies is the presence of endogenous β cat which may play a compensatory role. This has now been added to the discussion. “Additional complexity may also be due to the compensatory and/or competitive roles of the endogenous β cat allele in the NCAT/MCAT Crainbow models.” Current experiments, also beyond the scope of the manuscript, are underway to delete endogenous β cat (Floxed exons 2-6) in the background of MCAT or NCAT mice.*

2) Field cancerization involves somatic mutations that impact the competitive fitness of the cell involved. As these are rare events, the fitness bias that is of importance lies between the mutant cell and its normal neighbours. For the intestinal epithelium that is characterized by high-turnover, the central players are the stem cells that reside at the bottom of crypts. Therefore, mapping clonal competition involves two stages. First, the competitive bias of the mutant stem cell towards neighboring stem cells (scoring should be the fraction of the stem cell compartment within a crypt that has been taken over by the mutant, see many papers by the Winton lab). Second, upon crypt fixation, the pace of crypt fission (accelerated, neutral, etc) needs to be characterized as this underlies the mechanism by which mutant tissue can spread (concerns the scoring of the number of fully fixed crypts that encompass 1 clone). Counting the number of cells within a clone is not informative, as most cells within a clone are rapidly turned-over in the intestine. The focus must be on the number of stem cells (fraction of partial/full crypts) that fuel a clone.

Response 2. *Thank you for your thoughtful review of the two-step model of field cancerization. Throughout the study we accordingly assess field cancerization in two ways - crypt fixation and field spreading. Crypts that have been taken over by a mutant – i.e. Crypt fixation – reflects the potential of ISCs with somatic mutations to outcompete their neighbors and “drift” to monoclonality. This is similar to the scoring schema suggested and as previously utilized by the Winton lab and Clevers lab. In MCAT mice crypt fixation rates are relatively similar for each somatic mutation (Fig. 4f). Interestingly, crypt fixation was inhibited in ROBO*

mice, as shown by the increase in polyclonal crypts (Fig. 6). Field spreading values were also calculated to estimate how well somatic mutations spread through the intestine. We reasoned that crypt fission would increase field spread and result in the increase in the number of recombined epithelial cells. MCAT clones are rapidly “turned-over” and do not spread outside a single crypt. In contrast, ROBO clones spread laterally and do not appear to be “turned-over”. Quantifying total cells labelled is a simple and robust way to demonstrate this concept (Compare MCAT Fig4c to ROBO 7f).

3) Studying field cancerization using crainbow requires sporadic activation in adult epithelium. All experiments using constitutive Cre activities with onsets during embryogenesis (VillinCre) are not suitable for interpretation. First, (see point 1), I am questioning the oncogenic potential of the deltaN-beta catenin mutants. Second, there is no competition between mutant cells and normal neighbours. As all cells contain recombined crainbow alleles, there is only competition created between mutant clones. This has no relevance towards patients, as field cancerization are sporadic events. The only competitive bias that matters is between a mutant clone and normal neighbouring cells.

Response 3. Thank you for your suggestion and conceptually agree that large numbers of mutant clones are unlikely to be in competition with human patients. Nevertheless, competing fields to mechanistically evaluate related somatic mutations was one of our experimental objectives and led to a clinically relevant findings – namely, the acquisition of somatic mutations during perinatal development are able to rapidly spread. Previous work by Liskay and colleagues (PMID 23996931) demonstrated APC-loss during development is much more tumorigenic than APC-loss during adulthood. The well-known increase in crypt fission that occurs during postnatal growth of the intestine marks a critical period during which somatic mutations can more easily spread. Our findings demonstrate somatic field cancerization can occur during the first few years of life and suggests that large fields may be formed earlier than previously documented. This study provides proof-of-concept evidence for extensive field cancerization of the neonatal intestine.

4) Rosa-CreERT2 recombines the crainbow alleles in all cell types. Although most mutant cells are lost due to self-renewal of the epithelium, two types of clones resist. First, stem cell driven clones that are of interest to the study. The second however, are relatively long-lived (~up to 8wks) Paneth cells at crypt bottoms. These cells complicate analysis. As Paneth cells are post-mitotic, EdU pulses might be used to discriminate them from cycling stem cells. Alternatively (best option) is to use Lgr5-EGFP-Ires-CreERT2 knock-in mice to initiate sporadic clonal competition (crainbow activation) in stem cells.

Response 4. Thank you for the suggestions. We agree that Paneth cell hyperplasia within the crypt could confound the analysis. As shown in Fig5e (ROBO-Colon) we do not observe Paneth cell hyperplasia. In addition, we have provided extensive multiplex FISH analysis in Fig.5., illustrating that the somatic clones are not simply Paneth cells but rather include many of the cell types of the epithelial hierarchy (ISCs, TA cells, and enterocytes).

We have also included, in the response only, preliminary data using *Lgr5-EGFP-IRES-CreERT2* mice which confirm the *ROSA-CRE-ERT2* studies. Briefly, oncogenic *RSPO3* increases the number of

labelled crypts clonally expanding throughout the intestine. We conclude that this occurs due to an increase in the number of ISCs and is not consistent with an increase of Paneth cells or other terminally differentiated cell types. **(a)** 1 week post tamoxifen – vibratome slice and confocal imaging, **(b)** 8-weeks post tamoxifen – whole mount, **(c)** 18-weeks post tamoxifen – whole mount.

5) The work concerning *R-spondin* fusions is more interesting as it involves patient mutations. However, since these mutant cells establish neoplasias by themselves rather than pre-disposition, it disqualifies itself from a field cancerization study. Nevertheless, studying adenoma development driven by *R-spondin* fusions is still interesting by itself. Again, expression of *R-spondin* fusions from embryonic stages onwards is not interesting when it involves near-complete recombination of all cells in the epithelium. Adenoma development can be studied upon sporadic activation in adult epithelia as shown in the manuscript. However, clonal competition between different mutant clones is complicated in the case of *R-spondin*. Foremost, just like wild-type *R-spondin*, the described *R-spondin* fusions have been reported to be secreted. Therefore, the mutant phenotype is not intrinsic to the clone itself, but will also influence neighboring cells (potentially neighboring clones), severely complicating its analysis.

Response 5. Thank you for the helpful comments and suggestions. We also agree that the secretion of the *R-spondin*s can complicate the analysis. We have included a necessary caveat to the discussion. “*RSPO3* is a secreted protein and may have autocrine and paracrine effects on adjacent stem cells and the microenvironment.” As mentioned previously above, field cancerization can occur with or without morphological perturbations. Liskay and colleagues (PMID: 23996931) also found field cancer size is directly correlated to the adenoma risk. Their finding was in the context of a mouse model of *APC*-loss in which mice also eventually succumb to adenomas. Therefore, there is precedent in the literature for studying the progression of field cancers to adenomas.

Minor comments:

The word-art is very annoying. Please remove black outlines that surround coloured font.

Response. Apologies, we have reduced the line width of the outlines. We found the yellow color difficult to view in the absence of outlines.

REVIEWERS' COMMENTS:

Reviewer #3 (Remarks to the Author):

I accept the textual comments and nuances that have been incorporated in the current manuscript regarding the technical issues of the mouse data and interpretations raised in my report.

That said, I am still not convinced that 'field cancerization' is studied in the manuscript. For the main reason that field cancerization involves I) clonal expansion of mutant cells that II) predisposes to cancer development. Predisposition to cancer development has not been studied as the Bcat mice do not form adenomas, while the R-spo fusions are a driver by themselves.

Throughout the manuscript, including title and abstract, the wording 'field cancerization' needs to be replaced by 'clonal expansion'.

Reviewer #3 (Remarks to the Author):

I accept the textual comments and nuances that have been incorporated in the current manuscript regarding the technical issues of the mouse data and interpretations raised in my report. That said, I am still not convinced that 'field cancerization' is studied in the manuscript. For the main reason that field cancerization involves I) clonal expansion of mutant cells that II) predisposes to cancer development. Predisposition to cancer development has not been studied as the Bcat mice do not form adenomas, while the R-spo fusions are a driver by themselves. Throughout the manuscript, including title and abstract, the wording 'field cancerization' needs to be replaced by 'clonal expansion'.

Response. We appreciate the comments and feedback. We have replaced “field cancerization” with “clonal expansion” and derivatives (i.e. clonal spreading, oncogenic spread, etc.). In the introduction, we review pertinent field cancerization but we no longer conclude that Crainbow is a model of field cancerization. Rather we now highlight Crainbow as a useful tool to study attributes of field cancerization, for instance, stem cell competition and the spread of oncogenic clones.